# Drought decreases annual streamflow response to precipitation especially in arid regions

Alessia Matanó[1], Raed Hamed[1], Manuela I. Brunner[2,3,4], Marlies H. Barendrecht[5], Anne F. Van Loon[1]

[1] Institute for Environmental Studies, Vrije Universiteit Amsterdam, Amsterdam, The Netherlands

[2] Institute for Atmospheric and Climate Science, ETH Zurich, Zurich, Switzerland

[3] WSL Institute for Snow and Avalanche Research SLF, Swiss Federal Institute for Forest, Snow and Landscape Research WSL, Davos Dorf, Switzerland

[4] Climate Change, Extremes and Natural Hazards in Alpine Regions Research Center CERC, Davos Dorf, Switzerland

[5] Department of Geography, King's College London, London, United Kingdom

*Correspondence to*: Alessia Matanó (alessia.matano@vu.nl)

**Abstract.** Persistent drought conditions may alter catchment response to precipitation, both during and after the drought period, hindering accurate streamflow forecasting of high flows and floods. Yet, the influence of drought characteristics on the catchment response to precipitation remains unclear. In this study, we use a comprehensive dataset of global observations of streamflow and remotely sensed precipitation, soil moisture, total water storage and normalized difference vegetation index (NDVI). Using multivariate statistics on 4487 catchments with a stationary annual streamflow-to-precipitation ratio, we investigate the influence of drought on fluctuations of streamflow response to precipitation. Our analysis shows that generally droughts with streamflow or soil moisture anomalies below the 15th percentile lead to around 20% decrease in streamflow response to precipitation during drought compared to the historical norm. However, this decrease is reduced to only about 2% one year after the drought, which suggests a generally low influence of preceding drought conditions.. These effects are more pronounced when droughts are longer and more severe. Most changes were found in arid and warm-temperate regions, whereas snow-influenced regions exhibit less changes in catchment response due to drought. In addition, we used step-change analyses on 1107 catchments with non-stationary annual streamflow-to-precipitation ratio to identify significant abrupt shifts on the timeseries, examining the role of drought in driving these shifts. This analysis revealed both positive and negative shifts in annual streamflow response to precipitation after severe and persistent drought conditions regardless of climate and catchment characteristics. Positive shifts occur only when the drought propagated through the hydrological system after extended dry periods, while negative shifts are usually linked to shorter, intense dry periods. This study sheds light on the importance of considering climate characteristics in predicting dynamic catchment response to precipitation during and after persistent drought conditions.

## 1. Introduction

Drought is known to exert significant influence on catchment hydrological behaviour. Events such as the mega drought in Chile (Alvarez-Garreton et al. 2021; Garreaud et al. 2017), the millennium drought in Australia (Saft, Peel, Western, and Zhang 2016) and the 2011 Texas drought (Klockow et al. 2018) have resulted in substantial changes in vegetation productivity and type, soil hydraulic properties, surface water-groundwater interactions and water storage. Yet, understanding the extent of drought influence on catchment hydrologic response remains a crucial question with significant implications for enhancing hydrological prediction under future conditions.

Researchers have studied the impact of persistent drought conditions on catchment response using linear-regression approaches (Avanzi et al. 2020; Liu et al. 2022; Massari et al. 2022; Peterson et al. 2021; Saft et al. 2015; Saft, Peel, Western, and Zhang 2016; Wu et al. 2021) and water balance models (Liu et al. 2023; Maurer et al. 2022; Pan et al. 2020), registering a shift in rainfall-runoff relationships during long drought periods. According to Saft et al. (2015, 2016), persistent drought conditions in Australia's multi-year drought resulted in significantly less than expected runoff for some of the basins studied. This has been mainly attributed to reduced groundwater levels and hence, initial precipitation is used for replenishing water storage before runoff can occur. This process is prevalent in arid regions with high surface water-groundwater connection and large soil thickness, highlighting the linkage between changes in rainfall-runoff and catchment characteristics during persistent drought conditions. Peterson et al. (2021) have shown that rainfall-runoff shifts can persist after drought, in this case due to an increase in the fraction of precipitation going to evapotranspiration. Similarly to the Australian study, Avanzi et al. (2020) and Maurer et al. (2022) have identified less runoff during droughts in California than expected, attributing this to nonlinear feedback mechanisms between evapotranspiration and storage. Only a few catchments showed runoff increases mainly explained by catchment buffer capacities such as soil storage and snow-to-rain transitions.

Despite these findings, uncertainties remain on the specific catchment characteristics that contribute to vulnerability to drought-induced changes in the Q-P relationship, as well as the drought conditions that lead to these changes and the direction of the change (e.g., increase or decrease). Previous studies relied on samples with limited variability in catchment characteristics, with a large focus on natural catchments in Australia (Liu et al. 2021; Pan et al. 2020; Peterson et al. 2021; Saft et al. 2015; Saft, Peel, Western, and Zhang 2016) and California (Avanzi et al. 2020; Bales et al. 2018; Maurer et al. 2022). Furthermore, analyses of changes in rainfall-runoff relationships have primarily focused on the effects of meteorological droughts (Liu et al. 2021; Massari et al. 2022; Pan et al. 2020; Peterson et al. 2021; Saft et al. 2015; Saft, Peel, Western, and Zhang 2016), neglecting other drought types and failing to assess the effect of drought severity and duration on changes in the rainfall-runoff relationship.

Here, we analysed the temporal dynamics of the annual streamflow response to precipitation (computed as the ratio between annual streamflow and precipitation) in approximately 5000 catchments across the world. This annual Q-P ratio indicates the fraction of precipitation that is converted into streamflow within a year, providing insights into the catchment water balance. Specifically, we addressed the following questions: (1) how do drought characteristics (types, duration and severity) influence annual streamflow response to precipitation in general and in different hydro-climatic regions across the globe? and (2) when and where do abrupt changes in annual streamflow response to precipitation occur and how do those changes align with drought periods? To address these research questions, we first divided the catchments according to stationary and non-stationary streamflow-

precipitation ratio timeseries. Then, we employed mixed effects panel data models on stationary streamflow-precipitation timeseries to answer RQ1 and step-change analysis by using threshold regression models on non-stationary streamflow-precipitation timeseries to answer RQ2.

## 2. Methodology

### 2.1 Data preparation and drought detection

We identified a large sample of 5590 catchments, whose hydrometeorological timeseries span 25 to 34 years from 1980 to 2016. We compiled observed streamflow data from the Global Streamflow Indices and Metadata Archive (GSIM) database (Do et al. 2018a; Gudmundsson et al. 2018a). Using the catchment delineations in the GSIM dataset, we derived a set of hydro-climatic time series using Multi-Source Weighted-Ensemble Precipitation (MSWEP; Beck et al. 2019) for the precipitation sum over the catchment, the Global Land Evaporation Amsterdam Model (GLEAM; Martens et al. 2017) for surface (0 – 5 cm depth) and root zone (0 – 250 cm depth) soil moisture, the Gravity Recovery And Climate Experiment (GRACE; Boergens, Dobslaw, and Dill 2019) for total water storage, Landsat for surface water extent(Donchyts et al. 2016a; Earth Resources Observation and Science (EROS) Center 2022), and STAR - Global Vegetation Health Products for the normalized difference vegetation index (NDVI; NOAA 2022). These datasets and their post-processing are explained in more detail in Table 1 and in Matanó et al. (2024). For instance, for streamflow data, we included only GSIM stations with high delineation quality of their catchments, no missing months within a given year, and a minimum record length of 30 years.

**Table 1. Spatial resolution and temporal coverage of the data used in this study**

| Dataset | Temporal resolution | Spatial resolution | Temporal coverage |
|---|---|---|---|
| In-situ river streamflow data – GSIM(Do et al. 2018b; Gudmundsson et al. 2018b) | Daily statistics per month (MAX, MIN, MEAN) | nodes (catchment outlets/ hydrometric stations) | Varying (1900 - 2016) |
| Precipitation – MSWEP(Beck et al. 2019a) | monthly | 11 km | 1979 - 2022 |
| Soil Moisture - GLEAM (Global Land Evaporation Amsterdam Model) v3.6a(Martens et al. 2017) | monthly | 25 km | 1980 - 2020 |
| Surface water extent (Global Surface Water)(Donchyts et al. 2016b) | monthly | polygons/ centroids | 1984-2020 |
| No noise (smoothed) Normalized Difference Vegetation Index(NOAA 2022) | 7-day composite | 4km | 1981 - 2022 |
| Total Water Storage (TWS) anomaly is computed as standardized deviation of the | monthly | 50 km | 2002 - 2020 |

| GRACE satellite Liquid Water Equivalent (GRACE)(Boergens et al. 2019) | | | |
|---|---|---|---|

From average daily streamflow and total precipitation per month, we derived annual average daily streamflow (mm/day) and annual average daily precipitation (mm/day) for each catchment. As such, we assume that the storage change is negligible over an annual time scale. Data aggregation to a yearly scale was based on water years, defined for each catchment as the 12-month period beginning in the month of the lowest average monthly streamflow (Wasko, Nathan, and Peel 2020). We then applied a Box-Cox transformation (Sakia 1992) to normalize the skewed yearly streamflow distribution (Saft et al. 2015; Saft, Peel, Western, and Zhang 2016). This allowed us to obtain an approximately linear rainfall-runoff relationship, thereby facilitating the application of various statistical methods. Further, the Box–Cox transformation allowed us to overcome the issue of applying a log-transformation to streamflow timeseries with zero flow (e.g., ephemeral or intermittent rivers; Santos, Thirel, and Perrin 2018). We then computed annual streamflow-to-precipitation (Q-P) ratio timeseries for each catchment. This measure represents the annual runoff ratio and is dynamically influenced by climatic and landscape conditions. By considering an annual timescale, the ratio accounts for within-year evapotranspiration and storage processes within the catchment. It is important to note that, first, since the ratio is a lumped representation of these processes, it does not separate their individual contributions. Second, in some catchments, storage processes extend beyond a single year, which may influence the annual runoff ratio. This metric differs from other metrics such as elasticity (Anderson et al. 2023; Sankarasubramanian, Vogel, and Limbrunner 2001; Zhang, Viglione, and Blöschl 2022). While the annual runoff ratio provides an average measure of how much precipitation contributes to streamflow in a given year, elasticity tells us how streamflow reacts to changes in precipitation (Schaake 1990).

Drought events were detected using a variable threshold-level approach for perennial rivers (Van Loon 2015) and, a combined threshold-level and consecutive dry period method for ephemeral rivers (Van Huijgevoort et al. 2012; we refer to Matanó et al. 2024, for details on the method used for drought detection). We employed monthly-varying exceedance probabilities of the 15th, 10th, 5th, and 1st percentiles on precipitation, soil moisture, streamflow, total water storage, and surface water extent monthly timeseries. Additionally, NDVI anomalies per catchment were analysed to understand vegetation health and water flux dynamics. Drought characteristics were summarized at a yearly scale, by calculating maximum severity (defined as the difference between observed values and a predefined threshold), maximum cumulative severity (sum of consecutive severity across years), sum of severity, maximum cumulative duration (defined as the number of consecutive months in which observations are under a certain threshold), and sum of months under drought for each water year. These metrics were computed for each variable and were also aggregated for three types of drought: meteorological drought (based solely on precipitation data), soil moisture drought (incorporating surface and root zone soil moisture), and hydrological drought (taking into account streamflow, surface water extent, and total water storage).

## 2.2 Stationarity test and research framework

We tested the stationarity over time of yearly streamflow-to-precipitation ratios (Q-P) using the Augmented Dickey-Fuller (ADF) test (Paparoditis and Politis 2018), with a significance level set at 0.05. This test primarily

assesses whether the mean of the streamflow-precipitation relationship remained consistent over time, regardless of fluctuations around it.

We then divided our catchments in two groups: catchments with stationary Q-P timeseries and those with non-stationary Q-P time series. For stationary Q-P timeseries (ADF test p_value < 0.05), we evaluated the influence of drought on streamflow response to precipitation by employing a mixed-effects panel data model (Gelman and Hill

2007; Figure 1a). For Q-P timeseries displaying non-stationary behaviour (ADF test p_value > 0.05), we identified potential step-changes in the streamflow-to-precipitation ratio and their coincidence with drought conditions (Figure 1b). With the use of these two different approaches, we analysed both the dynamic influence of drought

on stationary Q-P time series (RQ1) and the more structural changes during drought in non-stationary series (RQ2).

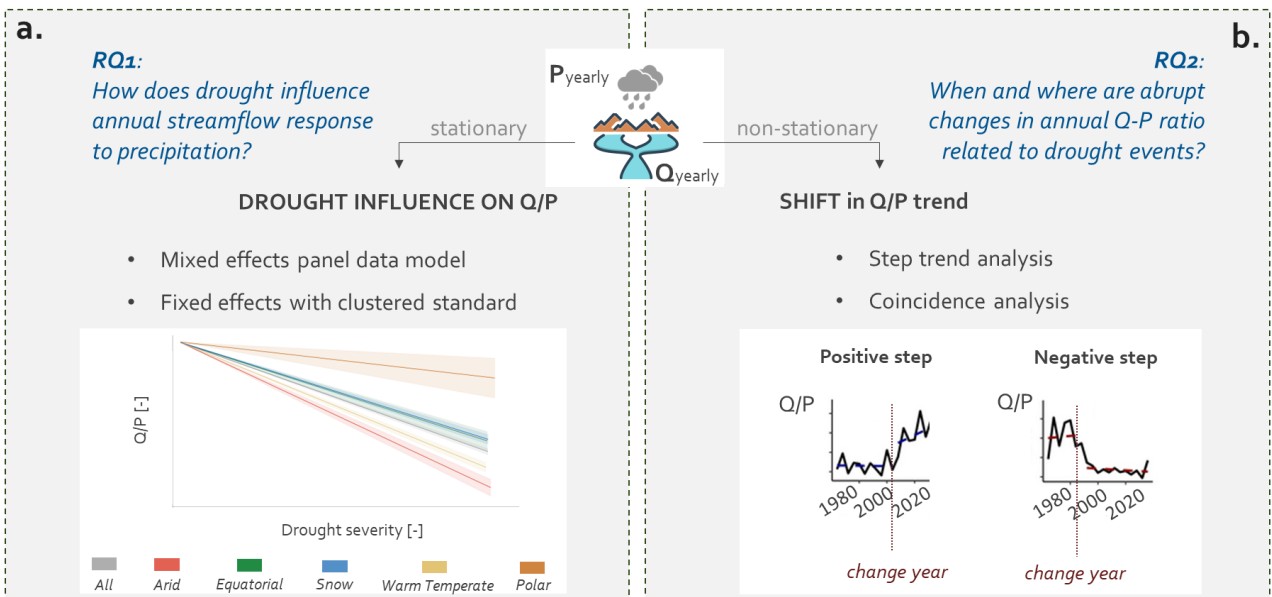

**Figure 1**: Research framework. (a) Methodology applied to investigate the influence of drought on annual streamflow response to precipitation. (b) Methodology applied to identify step changes in the Q-P ratio trend and system state conditions (e.g., anomaly presence) during the change year.

### 2.3 Panel data models for stationary Q-P timeseries

We used a mixed-effects panel data model (Gelman and Hill 2007) on 4487 catchments with a stationary

streamflow-to-precipitation ratio to explore the influence of drought conditions on the variability of annual streamflow response to precipitation over time. The mixed-effects model offers several advantages. First, it estimates both the general effect of drought characteristics on streamflow response to precipitation across all

catchments (fixed effect) and the variation of this effect between catchments (random effect). Second, this model is ideal for analysing hydrological units, as it can account for potential correlations between "nested" basins (Levy et al. 2018).

We ran two mixed-effects panel data models whose formulation is presented in Equations 1 and 4, to assess the impact of various drought types on the streamflow-to-precipitation ratio, accounting for different data availability.

Before employing these formulations, we tested several drought metrics such as maximum cumulative drought severity and maximum duration. However, as no substantial difference was found (see Supplementary Table S1), we opted for using maximum drought severity as a predictor for subsequent analyses.

In the first panel data model (Eq. 1, 2 and 3), we quantified the relationship between the variability of streamflow-to-precipitation and maximum drought severity for meteorological, soil moisture, hydrological droughts and NDVI anomalies. We considered the full length of the available timeseries (1982 to 2016) and the influence of drought severity in the same year and the year after (i.e., t and t-1 in Equation 1). Drought type variables included in the model formulation were selected based on correlation analysis (Figure S1 in Supplementary Information) and the length of their timeseries. For soil moisture drought, we used the maximum anomaly severity between soil moisture surface and root, given the high correlation between these anomalies (Figure S1c). Hydrological drought combining streamflow, surface water extent, and total water storage was chosen over the individual variables, as the latter two overlap with the other variables only for the last 10 years of data. Thus, hydrological drought was computed as the maximum anomaly severity among streamflow, surface water extent, and total water storage.

$$\left(\frac{Q}{P}\right)_{ct} = (\alpha + \alpha_c) + \sum_i^p (\beta_i + \beta_{ic}) * D_{i_{(t)}} + \sum_i^p (\gamma_i + \gamma_{ic}) * D_{i_{(t-1)}} + \varepsilon \qquad (1)$$

$$\sum_i^p D_{i_{(t)}} = D_{M_{sv\,(t)}} + D_{SM_{sv\,(t)}} + D_{HY_{sv\,(t)}} + D_{NDVI_{sv\,(t)}} \qquad (2)$$

$$\sum_i^p D_{i_{(t-1)}} = D_{M_{sv\,(t-1)}} + D_{SM_{sv\,(t-1)}} + D_{HY_{sv\,(t-1)}} + D_{NDVI_{sv\,(t-1)}} \qquad (3)$$

Where:

$c$ is a catchment index and $t$ is for year;

$\left(\frac{Q}{P}\right)_{ct}$: Ratio between annual average streamflow [mm/d] and precipitation [mm/d] calculated for the year $t$ in catchment $c$;

$\alpha$: Intercept ($\alpha$ for the fixed effect and $\alpha_c$ for the catchment specific effect);

$D_{i\,(t)}$: Max drought severity in the year $t$ (M: meteorological; SM: soil moisture, HY: hydrological and NDVI anomalies);

$D_{i\,(t-1)}$: Max drought severity ($sv$) in the previous year (M: meteorological; SM: soil moisture, HY: hydrological and NDVI anomalies);

$\beta_i$: Unique effect of drought $i$ occurred in time $t$ on the streamflow-to-precipitation ratio;

$\beta_{ic}$: Unique effect of drought $i$ occurred in time $t$ on the streamflow-to-precipitation ratio for catchment c;

$\gamma_i$: Unique effect of drought $i$ occurred in time $t$-$1$ on the streamflow-to-precipitation ratio;

$\gamma_{ic}$: Unique effect of drought $i$ occurred in time $t$-$1$ on the streamflow-to-precipitation ratio for catchment c;

$\varepsilon$: Error term.

In the second panel data model (Eq. 4, 5), we quantified the same relationship but this time using all variables as predictors (meteorological, soil moisture, streamflow, surface water extent and total water storage and NDVI anomalies), starting from 2002 to encompass the last 14 years. This time span was chosen to ensure complete

overlap of the total water storage and surface water extent timeseries with the other variables analysed.

$$\left(\frac{Q}{P}\right)_{ct} = (\alpha + \alpha_c) + \sum_z^p (\beta_z + \beta_{zc}) * D_{z_{(t)}} + \varepsilon \tag{4}$$

$$\sum_z^p D_{z_{(t)}} = D_{M_{sv\ (t)}} + D_{SM_{sv\ (t)}} + D_{STR_{sv\ (t)}} + D_{SW_{sv\ (t)}} + D_{TWS_{sv\ (t)}} + D_{NDVI_{sv\ (t)}} \tag{5}$$

Where:

$c$ is a catchment index and $t$ is for year;

$\left(\frac{Q}{P}\right)_{tc}$: ratio between mean streamflow [mm/d] and precipitation [mm/d] calculated for the year t in catchment c;

$\alpha$: Intercept ($\alpha$ for the fixed effect and $\alpha_c$ for the catchment specific effect);

$D_{i_{(t)}}$: Max drought severity in the year t (M: meteorological; SM: soil moisture; STR: streamflow; SW: surface water extent; TWS: total water storage and NDVI anomalies);

$\beta_z$: Unique effect of drought $z$ occurred in time $t$ on the streamflow-to-precipitation ratio;

$\beta_{zc}$: Unique effect of drought $z$ occurred in time $t$ on the streamflow-to-precipitation ratio for catchment c;

$\varepsilon$: Error term.

We assessed possible correlations among the predictors using Pearson correlation analysis. In the first model, the highest correlation (0.16) is observed between soil moisture and hydrological drought (Figure S1-e in Supplementary Information). In the second model, the highest correlation (0.18) is found between streamflow

drought and soil moisture (Figure S1-d in Supplementary Information). Similar correlation values were obtained using Spearman correlation analysis, which accounts for non-linear relationships (Figure S2 in Supplementary Information). These correlations are assumed to not significantly influence the estimation of the coefficients.

Autocorrelation in the residuals leads to an incorrect estimation of the variance of the estimated regression coefficients, hence a possible overestimation of the test significance (Anderson 1954). Therefore, we applied the Durbin-Watson test (Bartels and Goodhew 1981) to check for possible autocorrelation between the residuals,

obtaining values between 1 and 2, indicating little to no autocorrelation. We also applied the fixed effects panel data model with clustered standard errors (Moody, 2017) by catchment to test the robustness of our results. By using clustered standard errors, we allow for the possibility of correlated errors within each catchment, while

assuming that errors are independent across different catchments. As the number of clusters grows, the cluster-

robust standard errors become consistent. In applying the fixed-effects panel data model, we used the same regressions as in Equation 1 and 4. We first constructed a panel model using all available catchments, which yielded results consistent with those of the mixed-effects panel data model. Subsequently, we grouped catchments according to climate types - such as arid, snow, warm temperate, and equatorial, aligning with the Köppen-Geiger climate classification (Rubel and Kottek, 2010) and we applied the model to each category. Finally, we categorized the catchments according to climate and soil types, as well as climate and land cover types. For the soil-based categorization, we utilized soil classifications derived from the fractions of sand, silt, and clay within each analysed catchment, as provided by the GSIM dataset (Do et al. 2018a; Gudmundsson et al. 2018a). The land cover types used in the second categorization - 'Forest', 'Shrubland', 'Grassland' and 'Agriculture' - were also sourced from the GSIM dataset, which uses the United Nation Classification System for 2015 (European Space Agency (ESA) 2017) and assigns the land cover type that occupies more than 50% of the catchment area. The application of the fixed-effects panel data model to different clusters allowed us to compare coefficients across various catchment characteristics, and analyse whether these characteristics might alter the drought influence on the Q-P relationship.

The coefficients associated with the independent variables are dimensionless and indicate the magnitude of change in the response of streamflow to precipitation for a one-standard change in each respective independent variable, while holding all other variables constant. Finally, we analysed the spread of these coefficient values with catchment characteristics: mean annual catchment precipitation, maximum altitude, population density, and artificial water storage. The mean annual precipitation was computed using precipitation time series extracted from the MSWEP dataset, while the other variables were obtained from the GSIM dataset (Do et al. 2018a), which provides various attributes of catchment characteristics.

### 2.4 Trend and step-change analysis for non-stationary Q-P timeseries

To identify shifts in the streamflow response to precipitation from one steady state to another, we carried out a trend analysis in 1107 catchments with non-stationary streamflow-to-precipitation (Q-P) ratio timeseries. These catchments also have no more than two years of missing data in their streamflow timeseries. This involved modelling the relationship between the Q-P ratio and year (adapting the methodology in Berdugo et al. 2022). In detail, we investigated whether the Q-P trends are linear (i.e., monotonic trends or no trends), curvilinear (with an acceleration or deceleration that makes the trend nonlinear), or abrupt (characterized by a sudden change maintained until the end of the time period under analysis). We applied linear and quadratic models to test for linearity and nonlinearity, respectively, and also assessed the fit without a trend. Additionally, we used a threshold regression approach to detect any abrupt changes in the Q-P relationship. This approach models the relationship between variables that change at a specific threshold (i.e., change point). When multiple state transitions occurred within the analysed period, the method identifies the candidate change point that maximizes the goodness-of-fit or minimizes the loss function.

To select the best fitted model for each trend, we compared the Akaike Information Criteria (AIC; Wagenmakers and Farrell 2004) values of each fit. AIC is based on the log-likelihood of a given fit. A lower value indicates a model that fits the data better, but candidates with AIC differences lower than two units usually are similarly good.

To account for potential uncertainty in classifying trends due to their noisy nature and the relatively short length of the timeseries, we bootstrapped each timeseries 100 times without replacement and compared the model results of each bootstrapped iteration. For each bootstrap, we increased the probability of selecting the least influential points using the distance-based Mahalanobis method (Berdugo et al. 2022; Liu et al. 2018). We then computed the number of times that each model was selected as best-fit out of the 100 bootstraps, to identify the best fitted shape for each trajectory. We used this percentage as a measure of confidence for the best-fit shape of each trajectory (hereafter called confidence value).

Given the sensitivity of step regressions to outliers, we implemented three criteria to increase confidence in detecting step trends. First, we discarded step trends where the change point fell within the first or last three years of the period of analysis. This ensured that abrupt changes were not falsely identified due to anomalous data points at the start and end of the timeseries and guaranteed that detected abrupt changes persisted for at least four years after the change, indicating a certain stability of the change detected.

Second, we recorded the change point position (i.e., the year in which the trend is detected to change abruptly) for each trajectory classified as 'step-change' and calculated the mean and standard deviation (SD) of these change points across the 100 bootstrap iterations of each catchment. To determine the value of the change point SD that is critically influencing anomalous steps, we related the confidence value in the bootstrap selection and the SD of change points in all sites. We found that both parameters were related: for Q-P ratio timeseries in which the SD of change point was lower than 6 years, there was a strong negative correlation between the SD of change point and the confidence value in the bootstrap selection, whereas higher SDs in change point showed similarly low confidence values (Figure S3 In Supplementary Information). Therefore, we only considered step changes with standard deviations below 6 and confidence values above 80%.

Third, we categorized trajectories as step-change only if the Q-P ratios before and after the change point significantly differed according to a two-sample Kolmogorov-Smirnov test. This criterion ensured that the observed change point was sufficiently robust, aligning with the definition of a regime shift, characterized by significant differences in functioning or structure between two states. The analysis was carried out with a significance level of 0.05. For each of the trajectories classified as step-change, we identified the direction of the step as positive (increasing trend) or negative (decreasing trend). We then continued our analysis only considering the catchments with a step change in the Q-P timeseries.

For the trends classified as 'step-change', we examined drought anomalies occurring during and before the identified change years. Drought severity was categorized as moderate (between the 15th and 10th percentiles), severe (between the 10th and 5th percentiles), and extremely severe (below the 5th percentile).

## 3. Results

### 3.1 Drought influence on streamflow response to precipitation for stationary catchments

Generally, droughts tend to decrease the yearly response of streamflow to precipitation (negative coefficient values in Figure 2a and Tables S1 to S3), with hydrological drought having a more pronounced effect compared to other drought types. Soil moisture drought is the second most predominant factor (Figure 2a). In contrast, negative NDVI anomalies exhibit a slight increase (3%) in streamflow response to precipitation. The influence of drought persists into the following year, maintaining the same direction (in terms of increased or decreased catchment response to

different drought types) but with a reduced magnitude. Further, both drought severity and duration show a similar influence on streamflow response to precipitation (Table S1), likely due to the moderate correlation between these two variables.

While on average we find reduced streamflow response to precipitation during drought events, spatial variations among catchments exist (Figure 2b and Figure 3). In most climate zones, hydrological drought has the strongest influence on the Q-P relationship (Figure 2b). Arid regions are an exception, with soil moisture drought having the strongest influence on the Q-P relationship (a one-standard deviation increase in soil moisture drought severity leads to a 30% decrease in the Q-P ratio; Figure 2b). Further, NDVI anomalies in arid regions lead to a decrease in catchment response (a one-standard deviation increase in NDVI anomalies leads to a 3% decrease in the Q-P ratio) compared to the slight increase (around 5%) in response found in the other climate regions. Catchments located mainly in polar, snow-influenced and equatorial regions present the lower coefficient values, indicating less changes in streamflow response to precipitation during drought events (Figure 2b and Table S4). These findings are further supported by the random effects model, which identifies catchments with lower coefficient values in the Apennine region, southwest Canada, northeast United States, and central Brazil (Figure 3 and Figure S4 of the Supplementary Information).

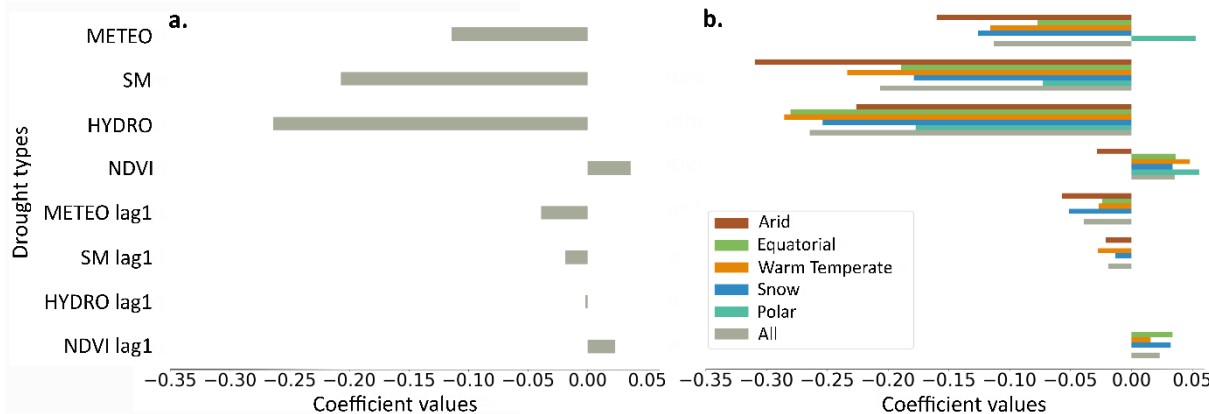

**Figure 2:** Bar plots of the panel data models' coefficient values for each drought type variable (METEO: meteorological, SM: soil moisture, HYDRO: hydrological and NDVI anomalies) with and without a lag time of 1 year. (a) Fixed effect coefficients from the mixed-effects panel data model. (b) Fixed effect coefficients from the panel data model with clustered standard errors, including all data and data grouped by climate types (refer to Table S3). All results are significant with p-values < 0.001, while results marked with asterisks indicate levels of significance: * p < 0.1 and ** p < 0.01. Missing bars indicate coefficients with p-values > 0.1, which are reported as NaN.

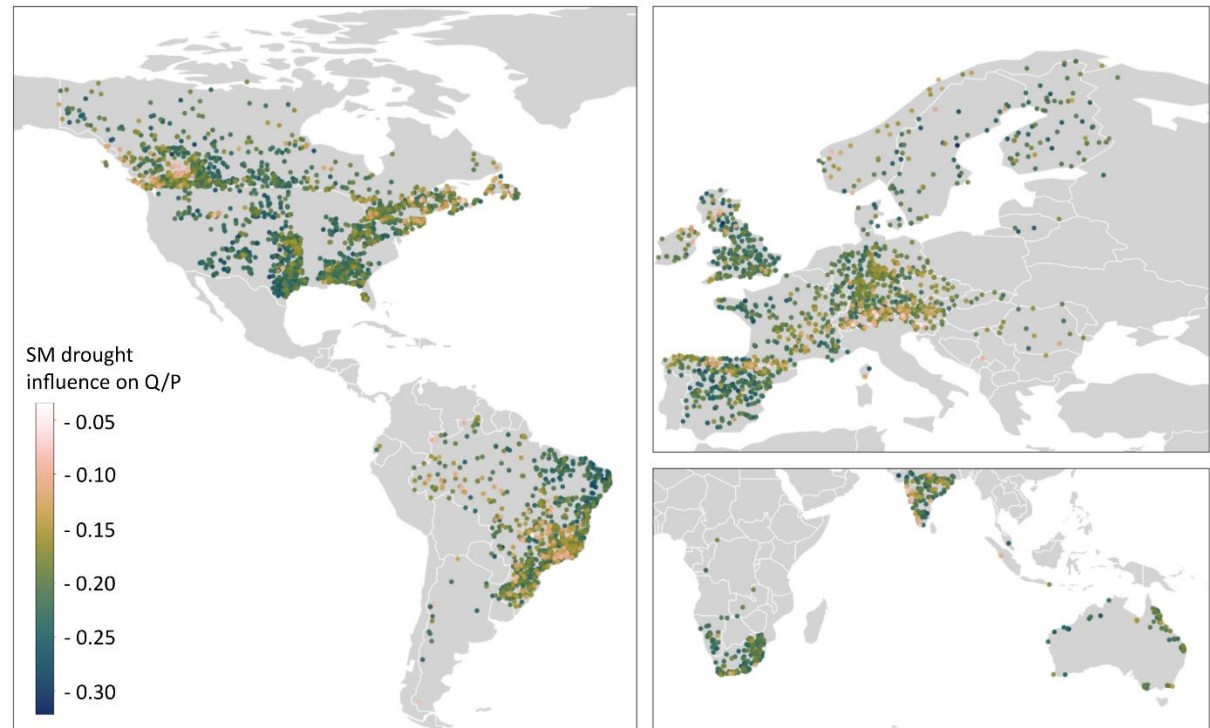

**Figure 3**: Catchment-specific effects of soil moisture (SM) drought on the Q-P ratio captured with the mixed-effects panel data model. Results are shown only for soil moisture as it exhibits the largest spatial variation compared to other drought types, which are reported in Supplementary Figure S3.

By identifying the dominant drought type, indicated by the highest regression coefficient value in each catchment, we determined which drought type primarily influences the Q-P relationship spatially (Figure 4a). This analysis also allowed us to assess the degree of catchment resilience to Q-P changes during droughts. Hydrological and soil moisture drought emerge as the most influential drought type (respectively for 30 and 27% of the catchments and indicated in brown and green in Figure 4), predominantly dampening the response of Q to P (Figure 4b). Soil moisture drought dominates in catchments clustered in the southcentral United States, southern Spain, and northeast India. On the other hand, anomalies in total water storage and NDVI affect Q-P relationships in 19% of the catchments each, with total water storage anomalies mainly in snow-influenced regions and northern Australia. Catchments with the highest regression coefficients (absolute values above 0.7) and indicating the lowest resilience to drought influence on Q-P relationships, are located in north Australia and the south-central/eastern United States and are primarily influenced by soil moisture and groundwater drought. The most resilient catchments (absolute coefficient values below 0.2) are found in the Alpine region and in southeast Brazil.

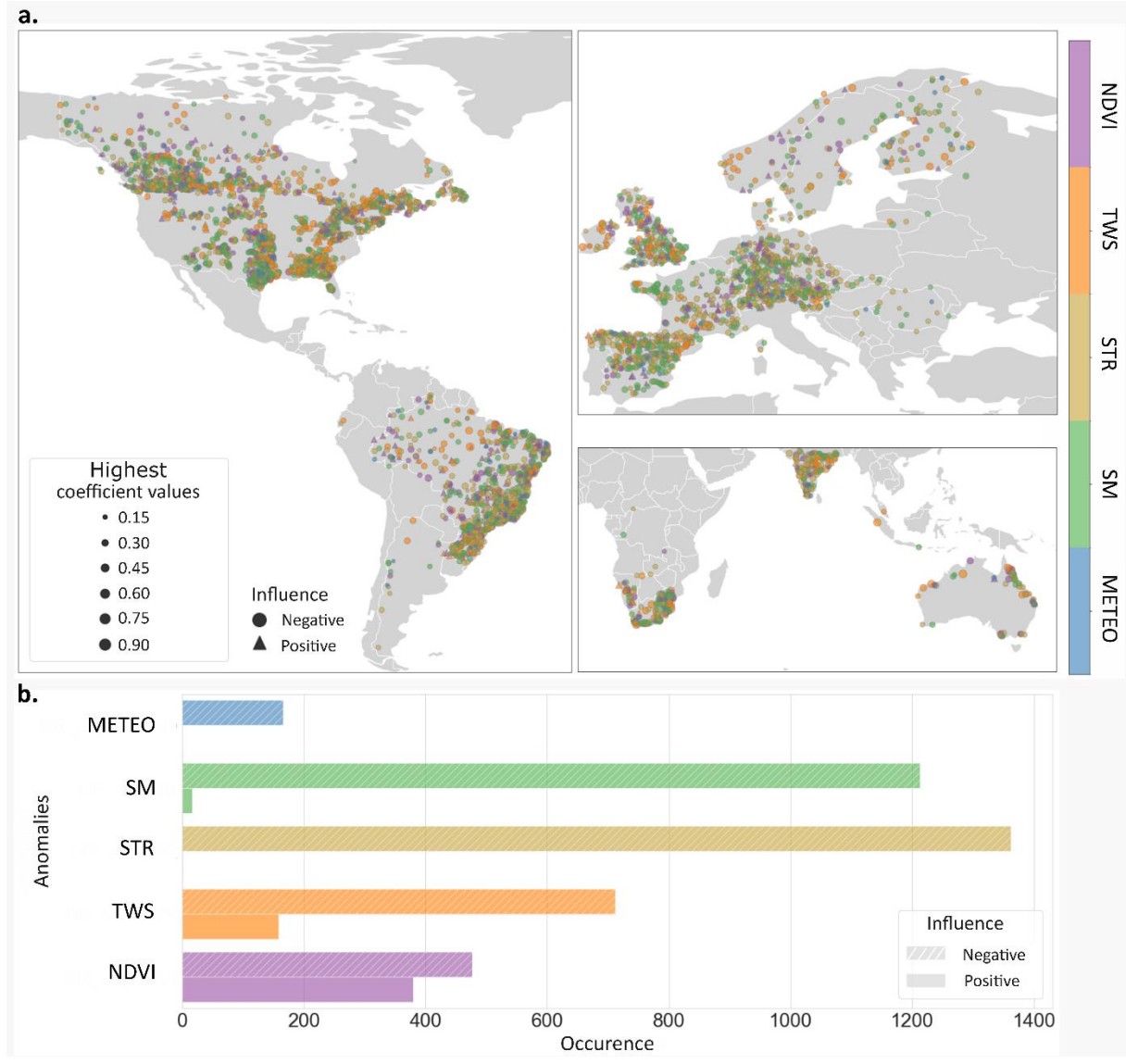

**Figure 4**: Highest regression coefficient per catchment, indicating the predominant drought type (among meteorological (METEO), soil moisture (SM), streamflow (STR), total water storage (TWS), and NDVI) influencing streamflow response to precipitation, as determined by the mixed-effects panel data model with time series data spanning the last 14 years (starting from 2002) and using Equation 2. This timeframe enables a full overlap of GRACE data with other variables. Coefficients of drought anomalies in surface water extent were excluded from the analysis due to nonsignificant results ($p > 0.1$). (a) Spatial distribution of the predominant drought type per catchment. Marker size corresponds to the magnitude of the highest coefficient. Circular markers represent a decrease in streamflow response to precipitation, while triangle markers indicate an increase in response. (b) Fraction of catchments with positive and negative coefficients of the predominant drought type per climate zone.

Spatial variations in streamflow response to precipitation due to drought are influenced by both topography and climate characteristics, but to different degrees. Altitude variation has a minimal effect on the influence of drought on streamflow response to precipitation (Figure S5 in Supplementary Information). As maximum catchment altitude increases, the sensitivity of catchment response to meteorological and soil moisture drought slightly decreases across all climate regions except arid ones, with this effect being particularly noticeable in the Alps, Pyrenees, mountain ranges of Norway, and the Canadian coastal mountains. In contrast, mean catchment precipitation exhibits a more pronounced effect, with a decrease in the drought influence on the Q-P ratio as mean

catchment precipitation increases (Figure S6 in Supplementary Information). The exception to this is hydrological drought, whose influence on streamflow response to precipitation slightly increases when mean catchment precipitation increases.

While climate types primarily influence variations in drought impacts on Q-P relationships across catchments, predominant land cover also plays a significant role (Supplementary Figure S8). Catchments dominated by grasslands and shrublands are more sensitive to Q-P changes induced by soil moisture drought, whereas those with forests and agricultural areas exhibit greater fluctuations in Q-P relationships during hydrological drought (first row of the heatmap in Supplementary Figure S8). These differences become more pronounced when catchments are clustered by both climate and land cover (Supplementary Figure S8). Specifically, grasslands in arid and equatorial regions exhibit heightened susceptibility to Q-P changes during drought. In snow-influenced climates, shrublands experience the most significant changes, while in warm temperate regions, agricultural and forested areas are the most affected. Conversely, negative NDVI anomalies have a minimal effect on the Q-P relationship in catchments dominated by grasslands.

Clustering catchments based on soil and climate type reveals that those in both snow-influenced regions and with sandy soils (sand fraction >33%) exhibit the least changes in streamflow response to precipitation due to drought (Supplementary Figure S9). Q-P ratios in arid and equatorial sandy catchments are significantly influenced by soil moisture drought, while hydrological drought plays a key role in warm temperate catchments with both clay and sandy soils. By clustering the catchments according to the total storage of the dams within a catchment, we can see that the influence of drought on the Q-P relationships slightly increases with an increase of reservoir storage (Supplementary Figure S11).

### 3.2 Analysis of step change in Q-P relationship for non-stationary catchments

The step analysis identified 197 catchments with a step change in the Q-P ratio timeseries, 183 of which occur during drought conditions. The percentage of catchments showing a step change was similar for both undisturbed and human-influenced (presence of reservoirs) catchments, at around 16%. Among the human-influenced catchments, 70% showed a negative step, whereas the undisturbed catchments were nearly evenly split, with about 52% exhibiting a positive step and 48% a negative step.

Catchment clusters with positive steps in the Q-P relationship (i.e., increased response of streamflow to precipitation) are primarily found in snow-influenced regions but are also present across other climate regions (Figure 5d). Those catchments are concentrated in the north-central United States, western Canada, and northeastern Brazil. Conversely, catchment clusters with negative step are found in southern Canada, scattered across Alpine and Scandinavian countries, and central Brazil. By plotting the years in which the steps occurred, we could identify some notable drought events (Figure 5). For instance, a cluster of catchments with negative step trend has the step change during the 2011-2012 drought that severely affected north-east Brazil (Rodrigues and McPhaden 2014). Within this cluster, only one catchment exhibits a positive step change. This catchment shares the same equatorial climate and has similar land cover as the others in the cluster (Table S5). The only notable difference is its significantly smaller size (hundreds of square kilometers compared to the others which span thousands).

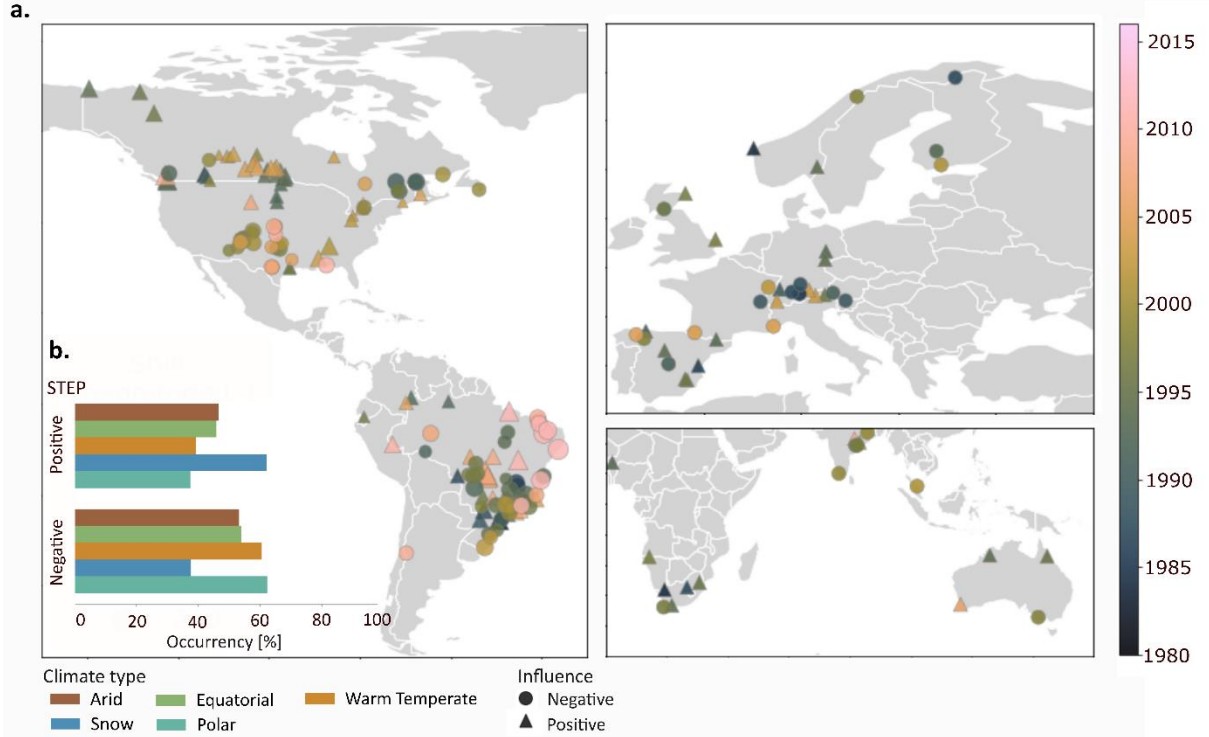

**Figure 5:** Global maps (a) of catchments whose Q-P ratio timeseries presents a step trend with a positive (triangle markers) or negative (circle markers) step. Marker colours indicate the years in which the step change occurred. Marker size indicate the

magnitude of the shift. b. Occurrence of positive or negative step change in Q-P relationship across catchments located in Arid, Warm temperature, Equatorial, Snow and Polar climate regions.

Drought events occurring during shifts in the Q-P relationship are typically extremely severe (below the 5th percentile; Figure 6a and b). This is especially pronounced in meteorological droughts and NDVI anomalies for negative shifts, and in soil moisture droughts for positive shifts. Our analysis of drought preceding the change year

reveals longer durations of soil moisture and hydrological drought (>1 year) for positive step trends, and longer durations of NDVI and meteorological droughts for negative steps (>10 months; Figure 6c and Supplementary Figure S13).

Finally, 96% of drought events detected during the change year had more than one anomaly, with 92% including meteorological droughts. Instances where the drought anomaly was solely meteorological resulted mainly in a decrease in the Q-P ratio following the step change (19% of catchments, Figure 7a). Conversely, instances showing

positive shifts were mainly related to at last two components of the hydrological system experiencing drought anomalies (Figure 7b). Specifically, both positive and negative shifts are initiated by precipitation anomalies, but the shift is positive mainly when this anomaly propagates to soil moisture (88% of catchments, Figure 7b) and

then to the hydrological system (75% of catchments, Figure 7b).

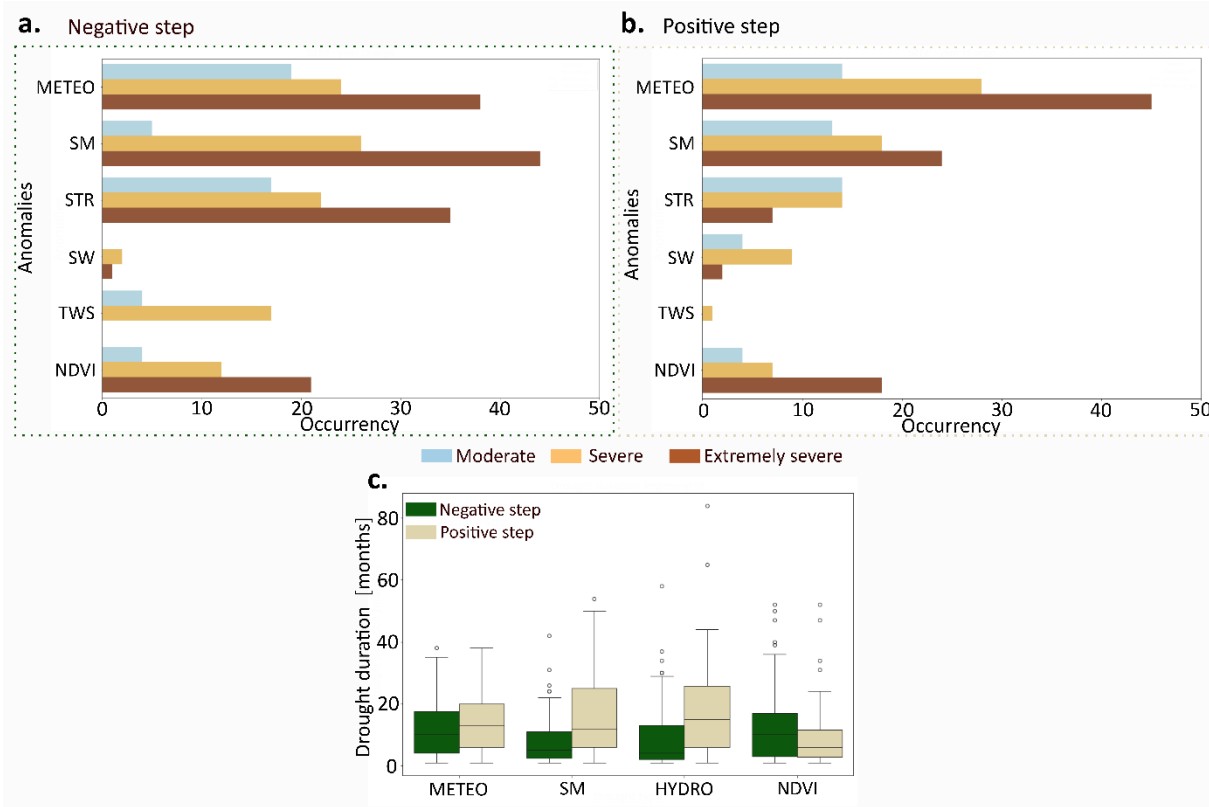

**Figure 6:** Analysis of severity and duration of drought events detected during the change years. (a-b) Occurrences of different drought types (meteorological (METEO), soil moisture (SM), streamflow (STR), surface water extent (SW), total water storage (TWS), and NDVI) for negative (a) and positive (b) steps. The fractions of total occurrences classified as moderate (10th < x < 15th), severe (5th < x < 10th), or extremely severe (x < 5th) droughts are represented by blue, yellow, and brown colours, respectively. (c) Total number of months under anomalies of consecutive drought years preceding the change year (drought may persist after the change year).

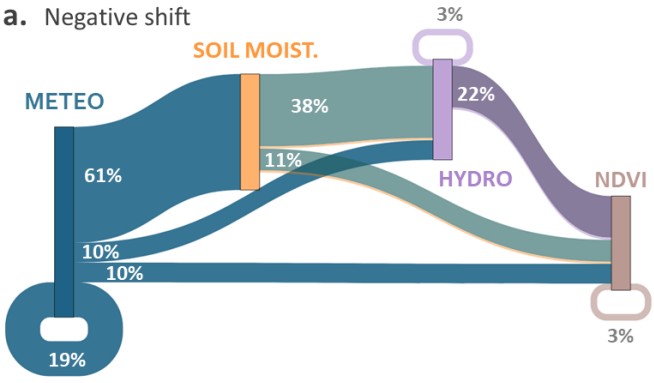

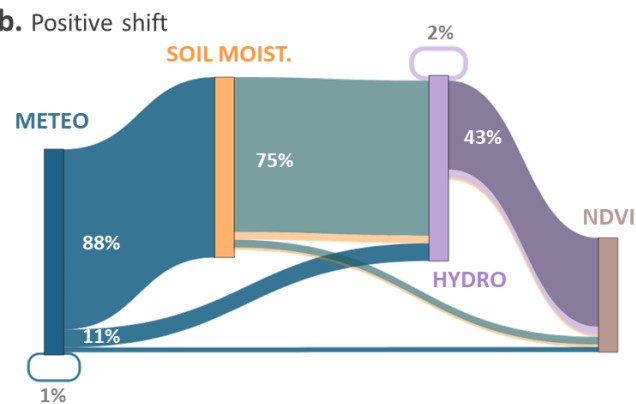

**Figure 7:** Percentage of drought types (co-)occurring during negative (a) and positive (b) shifts, propagating from precipitation, soil moisture, and streamflow droughts to further droughts along the drought propagation pathway. For example, in panel (b), the blue flow leading to the yellow bar (88%) indicates the co-occurrence of meteorological and soil moisture droughts, while the blue flow leading to the brown bar (11%) indicates the co-occurrence of meteorological drought and NDVI anomalies. Flow colours represent the co-occurrence of multiple anomalies (e.g., the green flow (75%) represents the co-occurrence of meteorological, soil moisture, and hydrological droughts). Circular flows (loops returning to the same drought type) represent events where only one anomaly was detected, indicating no further propagation of drought within the system. Percentages are calculated separately for positive and negative steps, representing the proportion of catchments exhibiting each specific co-occurrence relative to the total number of catchments showing a step change.

## 4.   Discussion

### 4.1 Drought influence on Q-P relationship in stationary catchments

The panel data analysis showed that drought in general decreases annual streamflow response to precipitation in stationary catchments (Figure 2 and Figure 3), aligning with previous research (Liu et al. 2021; Massari et al. 2022; Maurer et al. 2022; Saft, Peel, Western, Perraud, et al. 2016). This tendency can be explained by initial precipitation being used to replenish catchment water storage before streamflow responds (Barendrecht et al. 2024; Van Loon and Laaha 2015; Parry et al. 2016), which is further confirmed by the higher influence of hydrological and soil moisture drought on the Q-P relationship compared to meteorological drought and NDVI (Figure 2). On the other hand, negative NDVI anomalies lead to a slight increase of streamflow response to precipitation (Figure 2). This increase can be attributed to decreased evapotranspiration and reduced water uptake from dying vegetation (Breshears et al. 2005; Zhang et al. 2019). In smaller catchments (hundreds of square kilometers), an increase in

Q-P relationship may also be due to drought-induced soil compaction, which leads to reduced infiltration and
444 higher runoff (Alaoui et al. 2018; Descroix et al. 2009).

Despite a predominant tendency of decreasing streamflow response to precipitation during drought, the severity of this influence and the underlying processes differs spatially. Arid regions, for instance, show less resilience to
447 drought, which significantly influences catchment response to precipitation. This finding aligns with earlier studies (Liu et al. 2023; Maurer et al. 2022; Saft et al. 2015), which observed higher susceptibility to change in hydrological behaviour during persistent drought in arid catchments. Our study further reveals that Q-P
relationship in arid regions are particularly sensitive to soil moisture drought (Figure 3). This suggests that decreases in subsurface flow, which affect vegetation cover and surface water-groundwater interactions, are primary mechanisms driving reduced streamflow response to rainfall. Conversely, snow-influenced and polar
regions are more resilient to drought-induced changes in the Q-P relationship (Figure 2b) due to their high storage capacity. In these basins, snowmelt during drought can replenish subsurface storage, compensating for reduced precipitation inputs and limiting the dependency of evapotranspiration on deep subsurface storage (Avanzi et al.
2020). In these regions, the relationship between precipitation and streamflow is strongly influenced by drought anomalies in the total water storage (Figure 4), as confirmed by (Berghuijs and Slater 2023; Carroll et al. 2024; van Tiel et al. 2024), who highlight the importance of groundwater for mountain streamflow.

Spatial differences can also be found in the influence of negative NDVI anomalies on the Q-P relationship, though the overall influence remains small (less than 5%). While the yearly response of streamflow to precipitation generally increases during negative NDVI anomalies, in arid and semi-arid catchments, this response decreases
(Figure 2b). This decrease could partially be explained by reduced hydrological connectivity among bare patches (Jaeger, Olden, and Pelland 2014) (Urgeghe et al. 2010)and increased soil evaporation (Guardiola-Claramonte et al. 2011). However, these processes are highly dependent on the type, timing, and duration of drought, as well as
catchment-specific characteristics (Goodwell et al. 2018; Liu et al. 2024), making generalizations challenging. Furthermore, we acknowledge that reduced transpiration, typically associated with negative NDVI anomalies, may also influence the relationship (Johnson, Sinclair, and Kenworthy 2009).

Spatial variations are also driven by topographic characteristics and landcover type, although climate characteristics appear to be more predominant. In general, soil moisture and meteorological drought have a slightly smaller influence on streamflow response to precipitation at higher altitudes, with this behaviour accentuated
mainly in certain areas such as the Alps and Pyrenees. The same effect was found by Maurer et al. (2022) and explained by the resilience of high-elevation runoff to increases in potential evapotranspiration due to overall lower temperatures and sparser vegetation, (Garreaud et al. 2017) which help mitigate runoff losses elsewhere in the
basin. By analysing land cover, we find that forests reduce the influence of meteorological drought on catchment response, likely due to their higher hydraulic diversity, which buffers precipitation anomalies (Anderegg et al. 2018). However, when drought affects the hydrological system, forests present marked changes in catchment
response. A similar effect is observed in agricultural and grassland catchments, but specifically in response to soil moisture drought. It is important to consider that hydrological resilience to drought also varies with plant water use efficiency, which can lead to deviations from the general pattern observed (Xue et al. 2020).

While the impact of human influence (i.e., reservoirs) on drought-induced changes on the Q-P relationship is relatively weak, average catchment wetness, represented by mean annual precipitation, appears to have a stronger influence. In detail, we found a substantial decrease of soil moisture drought influence on the Q-P relationship with an increase in wetness which could be explained by the buffering effects of water storage (Liu et al. 2022).

### 4.2 Q-P shifts during drought in non-stationary catchments

The analysis of step changes in the Q-P relationship in non-stationary catchments showed slightly different patterns in how annual streamflow response to precipitation shifts during drought conditions, compared to Q-P fluctuations during drought in stationary catchments. While the study of drought influence primarily indicated a drought-induced decrease in streamflow response to precipitation, the step-change analysis identified both positive and negative shifts (Figure 5). These shifts occurred in various climate regions and under different catchment characteristics. This suggests that catchments might experience changes in the rainfall-runoff relationship regardless of their predominant climate and catchment characteristics.

Although both positive and negative Q-P shifts are found in catchments in different climate regions, catchments in snow-influenced regions exhibited a slight tendency toward positive shifts. These consistent increases in streamflow response to precipitation for at least four years after the shift can be explained by permafrost thaw (Lamontagne-Hallé et al. 2018) and glacial melt (Fountain and Tangborn 1985; Lutz et al. 2014; Schaner et al. 2012). While these mechanisms can sustain increased streamflow response to precipitation, they are ultimately finite resources. As glaciers and permafrost deplete and precipitation increasingly falls as rain, streamflow will eventually reduce (Berghuijs, Woods, and Hrachowitz 2014).

Contrary to the drought influence on stationary Q-P relationships, the severity and duration of droughts play a critical role in shaping these step changes (Figure 6). Our analysis indicates that severe droughts especially with longer durations are often linked to positive step changes in the Q-P relationship. For instance, positive step changes are frequently preceded by extended periods of severe soil moisture and hydrological drought, reflecting how persistent drought anomalies in the hydrological system can lead to significant adjustments in catchment response. These adjustments can be related to drought-induced changes in soil hydraulic properties (Alaoui et al. 2018; Descroix et al. 2009), vegetation type (Adams et al. 2012), interaction between shallow groundwater tables and soil moisture (Barendrecht et al. 2024). Conversely, negative step changes can occur after shorter drought periods, often linked to meteorological droughts. This suggests that negative step changes might be associated with more abrupt climatic shifts rather than longer-term changes in hydrological processes. This is further confirmed by the observation that positive Q-P shifts occur only when anomalies propagate through the hydrological system, resulting in multiple detected anomalies. In contrast, negative shifts can be recorded with only a decline in rainfall (Figure 7).

While there are no significant differences between catchments with human influence and those that are undisturbed when analysing drought influence on Q-P fluctuations, more pronounced differences emerged when analysing Q-P shifts during drought. Shifts occur in both catchments with reservoirs and those that are undisturbed. However, negative shifts are prevalent in catchments with reservoirs. This trend may be attributed to changes in reservoir operational rules aimed at drought mitigation (Di Baldassarre et al. 2017). Since a shift in our analysis must persist

for at least four years to be considered significant, this suggests that drought events have a lasting impact on reservoir operational strategies. These findings indicate that drought not only alters the Q-P relationships due to changes in the hydrological system but also through changes in risk perception and adaptation responses.

### 4.3 Limitations and challenges

The methodology and data employed in this study comes with a few limitations and challenges.

Firstly, the precision of estimates in mixed-effects panel data models improves with longer time series, as they enable more accurate modelling of random effects and mitigate the influence of short-term noise. Similarly, trend analysis benefits from extended time series. However, increasing the length of the time series can reduce spatial

coverage by excluding some catchments. To balance long-term coverage with spatial representation, we opted for a minimum time span of 25 years for streamflow and precipitation data. This decision, coupled with strict data quality checks (detailed in Matanó et al., 2024), resulted in underrepresentation of regions such as Asia, Australia,

northern and central Africa, and the western United States in our analysis.

Another significant challenge was the absence of GRACE measurements before 2002, which resulted in missing total water storage (TWS) anomalies for earlier years. Additionally, the surface water extent time series began in

1984, three years later than other variables. This led to a trade-off between maximizing the length of the time series in the panel data model and ensuring full overlap of all variables. To address this, we computed a new variable, the hydrological anomaly, summarising the anomalies in streamflow, surface water extent, and TWS to ensure a

consistent time span with the other variables. Additionally, we ran the panel data models using the last 18 years of data to guarantee full overlap of the variables without aggregation. In addition to differences in temporal scale, satellite datasets also exhibit varying spatial performance. For instance, GRACE has been shown to perform well

in North America and India but demonstrates lower accuracy in Europe. Similarly, MSWEP tends to perform better in the U.S. (Beck et al. 2019b), Europe, South America and Australia (Beck et al. 2017) while exhibiting lower accuracy in Africa (Beck et al. 2017). However, since our analysis includes only a small fraction of catchments

from Africa, potential errors due to lower performance in that region have a limited impact on our global assessment.

Further, although drought is a continuum, with temporal connectivity between events (Van Loon et al. 2024), our

analysis treats droughts as independent events, summarizing their characteristics at a yearly scale to facilitate comparison with the yearly ratio of Q to P. We only partially accounted for drought connectivity by incorporating drought characteristics from the preceding year into our analysis. However, their influence was minimal (less than

5%), with meteorological drought showing a slightly higher influence compared to other drought types.

The accuracy of the percentage values representing the influence of a certain drought type on the yearly Q-P ratio is affected by uncertainties in precipitation and streamflow observations. Although these percentage values are not

exact due to observational uncertainties, the relative magnitudes provides meaningful information, allowing us to identify which drought types have the strongest influence on the Q-P ratio.

Finally, another challenge lies in bridging the 'scale gap' between drought events, which occur on an event time

scale, and the streamflow-precipitation ratios, which are computed on an annual time scale. To mitigate this, we

calculated various metrics to represent the characteristics of drought events on a yearly basis, attempting to reconcile these different temporal scales within our analysis.

## 5. Conclusion

This study used panel data models to examine the effects of drought type, duration, and severity on streamflow
response to precipitation, accounting for variations in climate types, altitudes, land cover and average precipitation levels. Our analysis generally revealed a decrease in streamflow response during droughts in stationary catchments, except in cases of negative NDVI anomalies, which slightly increased catchment response. Spatial variability was
evident, with arid and semi-arid regions showing lower resilience to drought-induced changes in the Q-P relationship, while wet catchments, such as those in snow-influenced climates, showed greater resilience due to their water-buffering mechanisms. This trend of reduced response intensified with longer and more severe
droughts, though the effects of duration and severity were similar in magnitude. Further analyses based on step-change methods in non-stationary catchments revealed both positive and negative shifts in catchment response. Specifically, longer and more severe droughts related to soil moisture and hydrology often resulted in positive
shifts in response, whereas shorter, more abrupt meteorological droughts were associated with negative shifts. These findings underscore the complexity of drought impacts on the Q-P relationship and highlight the importance of considering both drought characteristics and regional differences when evaluating streamflow responses.
Understanding changes in catchment response to precipitation is crucial for assessing the resilience and adaptability of catchments to drought, given its distinct roles in influencing flow regimes.

**Data and code availability**

All data used in this study come from secondary datasets which are publicly available at the time of publication. Data regarding streamflow data are available through the GSIM dataset at
579 https://doi.pangaea.de/10.1594/PANGAEA.887477 (Do et al. 2018a; Gudmundsson et al. 2018a). Precipitation data can be downloaded at https://www.gloh2o.org/mswep/ (Beck et al. 2019b). Data on drought events are openly available at the following URL/DOI: https://figshare.com/s/a06830fc5111bd7804ce (Matanó et al 2023). Python
and R code for the complete analysis are available at https://figshare.com/s/20227bda9e89aca586da.

**Author contributions.** The conceptualization and methodology were designed by AM, AFVL, RH, MIB and
585 MHB. Data curation, formal analysis, investigation, software handling, visualization, and writing (original draft preparation) were performed by AM. Supervision and validation were carried out by AFVL, RH, MIB and MHB. Writing (review and editing) was carried out by AFVL, RH, MIB, MHB and AM.

**Competing interests**

One of the (co-)authors is a member of the editorial board of HESS.

**Acknowledgements**

This research was funded by the European Union (ERC, PerfectSTORM, ERC-2020-StG 948601). Views and opinions expressed are however those of the authors only and do not necessarily reflect those of the European Union or the European Research Council Executive Agency. Neither the European Union nor the granting authority can be held responsible for them. The authors thank SURF ([www.surf.nl](www.surf.nl)) for the support in using the National Supercomputer Snellius. RH acknowledge funding from the European Union's Horizon 2020 research and innovation programme under Grant Agreement No. 101003469 (XAIDA). MIB acknowledges funding by the Swiss National Science foundation (project 200021_214907).

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
