# Peer review of "Drought decreases annual streamflow response to precipitation especially in arid regions"

_EGUsphere, 2024_

## Referee Comment (RC1)

Review of "Drought decreases streamflow response to precipitation especially in arid regions" by Matanó et al.

The manuscript by Matanó and co-workers focusses on the response of catchment streamflow to precipitation, and how this response is changed by drought conditions. The authors analyze data from a large number of catchments globally, that allows for general conclusions on catchment behavior during drought. I enjoyed reading the manuscript, not in the last place because some of the findings have potential implications for our understanding of catchment functioning. However as with many studies on catchment observations, analysis can be complicated by the fact that these are not made under controlled conditions. In my view, some of the main conclusions of the manuscript are subject to a wrong interpretation of sensitivity, and to a lack of consideration of the natural variability in P that might affect the interpretation of changes in Q/P. These issues are discussed in more detail below.

The authors' work focusses on "the Q-P relationship", however the authors do not clearly define whether this should be seen solely as a catchment property, solely as a property of weather/climate forcing, or perhaps as a combination of both. This is important, because much of the interpretation of the results depends on this definition. Based on the wording in the manuscript, it seems that the authors implicitly assume the Q-P relationship to be a catchment property, which might change during or following drought, where the effect of climate is removed because P is used for normalization. A first problem with focusing on the Q-P relationship without clearly defining it, is that this relation by itself is not an established concept. It is not generally considered to be a catchment property, nor is the concept used in any predictive hydrological model (for a good reason). Streamflow generally originates from storage, making it tricky to discuss a relationship between P and Q without directly involving S. In has been shown by numerous studies (perhaps most notably by Kirchner, WRR, 2009) that the S-Q relationship rather than the P-Q relationship is a useful predictive model and that the sensitivity of discharge to storage, but not the sensitivity of discharge to precipitation, can be seen as a (nonlinear) catchment property. This means that the (yearly) Q/P ratio will depend on more than just catchment characteristics, and given the importance of catchment storage and its relation to discharge, it might also depend on the value of P itself in spite of the normalization used. I have quickly checked this on data from a catchment with long-term observations, and found that indeed there seems to be a clear relation between Q/P and P itself:

[Figure]

The dependency of Q/P on P itself has two implications for the results presented in the manuscript. First, it undermines the interpretation of single Q/P values in terms of sensitivity. In my view, sensitivity is the slope of the relation between Q and P as illustrated in the graph below (based on the same data):

[Figure]

This relation might well be continuous and linear over drought and non-drought periods with the same effective sensitivity or slope dQ/dP (here 1.06, so effectively 1). This example also illustrate that the conceptual framework in Figure 1 contains an important error: a yearly observation of Q/P is not the same as the sensitivity of yearly discharge to P. The real sensitivity is rather constant over the observed range in spite of the ratio Q/P getting smaller under dry conditions. Secondly, it means that, following the principle of Occam's razor, changes in P should first be ruled out as a possible explanation for any of the conclusions (particularly when discussing shifts and non-stationarity). Only when the shifts cannot be explained by changes in P, it is meaningful to discuss potential other mechanisms such as mentioned in the discussion. In summary, while the manuscript deals with an interesting and relevant topic, it currently suffers from a lack of clear definitions and consistency. In order for the manuscript to become a significant contribution to the understanding of drought impact on streamflow, in my view the authors will need to: a) investigate the real sensitivity of Q to P (from a

linear fit as in the second figure) and to what extent drought years might differ from the relation as given by non-drought years, and b) exclude the possibility that shifts in Q-P behavior are attributed to catchment processes whereas in reality they are simply induced by changes in P (for instance due to circulation changes). I realize this will require additional analysis, but given the potential significance of the findings it is important that they are based on careful analysis of all factors involved, so including dependency of Q-P on P itself.

---

## Author Comment (AC1)

**Drought decreases streamflow response to precipitation especially in arid regions**

We would like to thank the reviewer for the time taken to review our manuscript. In the following pages, we respond to the comments of the reviewer. Our responses are shown in blue, the revised text is shown in *italics*, and line numbers mentioned in this response refer to the current version of the manuscript and they are indicated within brackets [xx].

**Reviewer 1**

The manuscript by Matanó and co-workers focusses on the response of catchment streamflow to precipitation, and how this response is changed by drought conditions. The authors analyze data from a large number of catchments globally, that allows for general conclusions on catchment behavior during drought. I enjoyed reading the manuscript, not in the last place because some of the findings have potential implications for our understanding of catchment functioning. However as with many studies on catchment observations, analysis can be complicated by the fact that these are not made under controlled conditions. In my view, some of the main conclusions of the manuscript are subject to a wrong interpretation of sensitivity, and to a lack of consideration of the natural variability in P that might affect the interpretation of changes in Q/P. These issues are discussed in more detail below.

- We thank the reviewer for taking time to read our manuscript. We are pleased that the reviewer found the topic interesting and recognized the potential significance of our findings. We also appreciate the reviewer's constructive comments. Particularly, the reviewer highlights the need for:

In summary, while the manuscript deals with an interesting and relevant topic, it currently suffers from a lack of clear definitions and consistency. In order for the manuscript to become a significant contribution to the understanding of drought impact on streamflow, in my view the authors will need to: a) investigate the real sensitivity of Q to P (from a linear fit as in the second figure) and to what extent drought years might differ from the relation as given by non-drought years, and b) exclude the possibility that shifts in Q-P behavior are attributed to catchment processes whereas in reality they are simply induced by changes in P (for instance due to circulation changes).

- We agree with the reviewer that a clear definition of the yearly Q-P ratio is essential for the interpretation of the results and we will further expand on it in the Introduction and Methodology sections.

    [74] Here, we analysed the temporal dynamics of the *annual* streamflow *response* to precipitation (computed as the ratio between annual streamflow and precipitation) in approximately 5000 catchments across the world. *The*

*annual Q-P ratio indicates the fraction of precipitation that is converted into streamflow, providing insights into the catchment's water balance.*

[109] We then computed yearly streamflow-to-precipitation (Q-P) ratio timeseries for each catchment. *This measure represents the annual runoff ratio and is dynamically influenced by climatic and hydrological conditions. By considering an annual timescale, the ratio inherently accounts for evapotranspiration and storage processes within the catchment. However, it is important to note that, first, since the ratio is a lumped representation of these processes, it does not separate their individual contributions. Second, in some catchments, storage processes extend beyond a single year, which may influence the annual runoff ratio.*

Regarding the first reviewer's point (a), we acknowledge that the term 'sensitivity' used in the manuscript is not appropriate, as it implies a proportional change in streamflow divided by the proportional change in precipitation (Sankarasubramanian et al., 2001; Schaake, 1990). Instead, we computed the yearly ratio of streamflow to precipitation. Our focus is indeed on understanding whether and how the yearly fraction of precipitation to streamflow is influenced by different drought types (e.g., anomalies in precipitation, soil moisture, or storage). Therefore, we will replace the term "sensitivity" with "response" throughout the manuscript, as this better reflects the measure we are using. Examples of these changes are as follows:

[76] Specifically, we addressed the following questions: (1) how do drought characteristics (types, duration and severity) influence *yearly* streamflow *response* to precipitation in general and in different hydro-climatic regions across the globe? and (2) when and where do abrupt changes in streamflow *response* to precipitation occur and how do those changes align with drought periods?

[145] We used a mixed-effects panel data model (Gelman and Hill 2007) on 4487 catchments with a stationary streamflow-to-precipitation ratio to explore the influence of drought conditions on the variability of *yearly* streamflow *response* to precipitation over time.

- Concerning the second reviewer's point (b), this is precisely the main objective of our study: to investigate the influence of different drought typologies (i.e., meteorological, soil moisture and hydrological drought) on the response of streamflow to precipitation. We recognize that the wording in the manuscript may have unintentionally suggested that the Q-P relationship is solely a catchment property. This was not our intention. We will revise the introduction and methodology to clarify that our study does not consider the Q-P relationship as an

inherent catchment property but rather as a dynamic relationship influenced by climatic and hydrological conditions (revision for instance suggested above for the definition of the Q/P ratio).

Further, the reviewer's analysis shows a significant correlation between Q/P and P. However, for the scope of our study, this does not pose a problem. On the contrary, it validates our approach of accounting for the effect of meteorological drought on the annual Q-P relationship. Indeed, in our study, we explicitly use precipitation anomalies as one of the predictive variables, ensuring that the effect of climate is accounted for.

Finally, we thank the reviewer for pointing out the issue with the conceptual framework in Figure 1. We agree that figure 1a might be misleading, as we cannot directly reconstruct the individual values of Q and P from their ratio. As we only analyse the relationship between these two variables (predicted variable), we could only plot how those relationship (Q/P) change in relation to drought and no-drought years. We will revise Figure 1 accordingly, ensuring it reflects the methodology used in our study.

**References**

Sankarasubramanian, A., Vogel, R. M., & Limbrunner, J. F. (2001). Climate elasticity of streamflow in the United States. *Water Resources Research*, *37*(6). https://doi.org/10.1029/2000WR900330

Schaake, J. C. (1990). From Climate to Flow. In *Climate Change and U.S. Water Resources*.

---

## Author Comment (AC2)

**Drought decreases streamflow response to precipitation especially in arid regions**

**Reviewer 3**

I found this is an interesting article. I have a few comments detailed below. Minor revision is requested.

- We thank the reviewer for taking time to read our manuscript and we are very pleased that they found the manuscript interesting. The reviewer provides constructive feedback and suggestions, which we will address in the revised manuscript. Below, we summarize the changes we will make in response to these comments. Our responses are shown in blue, the revised text is shown in *italics*, and line numbers mentioned in this response refer to the current version of the manuscript and they are indicated within brackets [xx].

1. Title: I wonder whether the article title chosen by the authors is clear enough. I found it very general and therefore not really convincing on the original results it brings. For example should the annual scale of the analysis be mentioned.

   - We thank the reviewer for the suggestion and we agreed about the importance of adding the temporal scale of the analysis in the title: Drought decreases *yearly* streamflow response to precipitation especially in arid regions

2. Abstract: Are the 2%-3% evolutions significant given all the other uncertainties in data?

   - If the reviewer is questioning the relevance of the findings showing 2–3% influence of NDVI anomalies and drought events from the previous year on the Q-P ratio, we argue that the broader conclusion remains valid despite uncertainties in the data. A relatively small influence suggests that this specific drought type has minimal impact on catchment response. In contrast to the 20–30% changes observed for other drought types, this lower effect may indicate that these catchments are more resilient to changes associated with NDVI. Furthermore, this indicates that the influence of preceding drought events appears to have minimal impact on the yearly Q-P ratio. We will further specify this in the abstract:

     [23-27] Our analysis shows that generally droughts with streamflow or soil moisture anomalies below the 15th percentile lead to around 20% decrease in streamflow sensitivity to precipitation during drought compared to the historical norm. *However, this decrease is reduced to only about 2% one year after the drought, highlighting the generally low influence of preceding drought conditions*. These effects are more pronounced when droughts are longer and more severe.

3. Introduction: The runoff-to-precipitation ratio was heavily analysed in studies based on the Budyko approach. I find it may be useful to more explicitly make a link with the studies which

analysed the sensitivity/elasticity of this approach to various variables and discuss how the proposed study can be linked to these previous works (e.g. Xue et al., 2020)

- We thank the reviewer for this suggestion and for highlighting the work by Xue et al. (2020). In line also with the other reviewers' comments, we agree on the need to better define the yearly Q-P ratio and compare it to other metrics used in the literature (e.g. elasticity). We will add the text below and further check for possible links with the elasticity metric computed through the Budyko Framework by (Creed et al., 2014; Helman et al., 2017; Xue et al., 2020).

  > [109] We then computed yearly streamflow-to-precipitation (Q-P) ratio timeseries for each catchment. *This measure represents the annual runoff ratio and is dynamically influenced by climatic and hydrological conditions. By considering an annual timescale, the ratio inherently accounts for evapotranspiration and storage processes within the catchment. However, it is important to note that, first, since the ratio is a lumped representation of these processes, it does not separate individual contributions. Second, in some catchments, storage processes extend beyond a single year, which may influence the annual runoff ratio. This metric differs from other metrics such as elasticity (Anderson et al., 2023; Sankarasubramanian et al., 2001; Zhang et al., 2022). While the annual runoff ratio provides an average measure of how much precipitation contributes to streamflow in a given year, elasticity tells us how streamflow reacts to changes in precipitation (Schaake, 1990).*

4. Section 2.1: I liked the fact that a large data set was used in this study. However I missed a discussion on data quality and possible dependency of results to the type of data used. For example, satellite products are known to be subject to large biases, which are not uniform whatever the regions or conditions. Besides they often show non stationary behaviour over time due to changes in algorithms or data. How these uncertainties may impact results shown in this study? A more detailed description of data used on these aspects would be useful.

   - We agree with the reviewer about the inhomogeneous spatial and temporal performance of global satellite data. In accordance with this, we explored below possible spatial and temporal differences in biases and accordingly we will add these as limitations in the revised manuscript. In particular, we will focus on MSWEP, GRACE and NOAA-NDVI, as these are satellite-based products. In contrast, GSIM relies on observations, and GLEAM soil moisture is modelled data. The limitations of these latter datasets are discussed in Matanó et al. (2024), and we have linked these discussions to our manuscript:
   [97] These datasets and their post-processing are explained in more detail in Table S1 of the Supplementary Information and in Matanó et al. (2024).

According to studies that compared satellite precipitation datasets (Gebrechorkos et al., 2024; Mazzoleni et al., 2019), there is no single best-performing precipitation dataset for all regions, and the performance is sensitive to basin characteristics. However, several studies (e.g., (Beck et al., 2017; Satgé et al., 2019) have showed MSWEP's strong spatial performance compared to other datasets, such as ERA5 and CHIRPS, across various global regions. That said, MSWEP tends to perform better in the US, South America, Australia and Europe (Beck et al., 2017, 2019) while exhibiting lower accuracy in Africa (Beck et al., 2017). However, in our study, only a small fraction of stations in Africa passed the quality check, making their contribution to the total dataset minimal. Therefore, spatial biases from MSWEP's performance are likely negligible in the context of our global analysis.

Concerning GRACE: validating the spatial and temporal quality of GRACE data is challenging due to the limited global coverage and the insufficient density of in situ measurements across all hydrological reservoirs (as discussed inSchmidt et al., 2008). However, some studies have attempted regional validation. For example, in South America, GRACE has demonstrated good performance in distinguishing hydrological signals from various reservoirs (Schmidt et al., 2008). Similarly, GRACE have shown strong agreement with local observations in reproducing groundwater storage anomalies at the basin scale in India (Bhanja et al., 2016) and in north America (Wang et al., 2022). In contrast, its performance in Europe has been reported to be lower (Van Loon et al., 2017).

Regarding the transition from GRACE to GRACE Follow-On (GRACE-FO), we note that this did not impact our analysis, as our study covers the period between 1980 and 2016, while GRACE-FO commenced in 2018.

Regarding NOAA-NDVI, we found only regional or country-level studies that validated its spatial reliability. For instance, studies in Australia (Holm et al., 2003) and East Africa (Nicholson et al., 1990) have shown significant performance of the NDVI dataset in capturing vegetation dynamics.

In the manuscript, we will acknowledge the potential uncertainties associated with satellite-derived data:

[567] In addition to differences in temporal scale, satellite datasets also exhibit varying spatial performance. For instance, GRACE has been shown to perform well in North America and India but demonstrates lower accuracy in Europe. Similarly, MSWEP tends to perform better in the U.S. (Beck et al., 2019), Europe, South America and Australia (Beck et al., 2017) while exhibiting lower accuracy in Africa (Beck et al., 2017). However, since our analysis includes only a small fraction of catchments from Africa, potential errors due to lower performance in that region have a limited impact on our global assessment.

5. Section 2.1: I found that Table S1 would be better placed in the main text of the article. This table is important to understand the variety between data used, e.g. in terms of periods available. I was also wondering which quality checks were done on the data used and how gaps in series were processed and accounted for in the models. If all catchments were plotted on a Budyko-type plot, could some specific/outlier behaviours be detected?

- We agree with the reviewer's suggestion and will move Table S1 to the main text. Regarding quality checks, we provided additional clarification in Matano et al. (2024) and Supplementary Note 1 of that paper but we will also add some of these clarifications in the main text. Specifically:
  - For streamflow data, we only included stations with high delineation quality for their catchments.
  - We analysed only stations with no missing months within a year and a minimum of 30 years of data.
  - For step-change analysis, stations with more than two years of gaps were excluded.

[97] These datasets and their post-processing are explained in more detail *in Table 1* and in Matanó et al. (2024). *For instance, for streamflow data, we included only GSIM stations with high delineation quality of their catchments, no missing months within a given year, and a minimum record length of 30 years*.

[253] To identify shifts in the streamflow response to precipitation from one steady state to another, we carried out a trend analysis in 1107 catchments with non-stationary streamflow-to-precipitation (Q-P) ratio timeseries. *These catchments also have no more than two years of gaps in their streamflow timeseries.*

Overall, fewer than 2% of the stations analysed had more than three years of total gaps in their time series. The strict criteria applied for data quality resulted in limited data coverage in regions like Asia, Australia, northern and central Africa, and the western United States, as also acknowledge in line 557 of the manuscript.

Regarding the Budyko-type plot, we also conducted this analysis for some catchments. Specifically, we applied the Budyko framework to analyse the catchments that exhibited a step change in the yearly Q-P ratio. We computed the ratio of actual evapotranspiration to precipitation, and the ratio of potential evapotranspiration to precipitation, both before (in black) and after (in red) the change year (see figure below). However, we decided to not include it in this manuscript. Adding this analysis would introduce another layer of results to a study that already implements two methodologies: the mixed-effect panel data model and the step-change analysis. So we decided to leave out this analysis and use it for a follow-up work / future paper.

[Figure]

6. Section 2.1: Could there be any influence of year-splitting on results, especially on the memory to drought conditions? The use of hydrological years make sense, but it will likely split drought events in two parts, i.e. straddling two years. I was also wondering if the hydrological year was determined catchment by catchment or if an homogeneity was sought between catchments in a same region or under similar climate type.

- We acknowledge the reviewer's concern that splitting drought events across water years could influence the results. While we recognize that drought events are continuous phenomena, we partially addressed this by considering the influence of drought conditions from the preceding year. Aggregating drought events into longer time periods could have reduced the number of events available for analysis, which would have further limited the robustness of our results. Further we have identified water years for each catchment as the 12-month period beginning in the month of the lowest average monthly streamflow (as reported in line 102 of the manuscript). In line with this, we will add to the text:

  [568] *Although drought is a continuum with temporal connectivity between events (Van Loon et al., 2024), our analysis treats droughts as independent events, summarizing their characteristics at a yearly scale to facilitate comparison with the yearly ratio of Q to P. We only partially accounted for drought connectivity by incorporating drought characteristics from the preceding year into our analysis. However, their influence was minimal (less than 5%), with meteorological drought showing a slightly higher influence compared to other drought types.*

7. Section 2.1: Which exponent values were used in the Box-Cox transformation?

- For the Box-Cox transformation, we used the Python function scipy.stats.boxcox. The exponent value for the transformation, defined as lambda (λ), determines the nature of the transformation. In our case, we set lambda to None, allowing the function to automatically estimate the optimal value of λ that maximizes the log-likelihood function.

8.  Section 4: I was not really convinced by several points in the discussion were the authors try to find explanations to the results found. These explanations remain hypotheses and should more clearly be presented as such.

> We agree with this point which was also raised by the second reviewer. In the discussion section, our intention was to assess whether our results align with findings from other studies and to explore how these findings have been explained in terms of underlying processes. Upon revisiting the references, we acknowledge that the use of Urgeghe et al., 2010, may indeed be an overextension in this context. As well as the use of Garreaud et al., 2017, so we will delete it. In agreement with the reviewer's comment, we will modify the discussion as following:

>> [485] Spatial differences can also be found in the influence of negative NDVI anomalies on the Q-P relationship, *though the overall influence remains small (less than 5%)*. While the response of the Q-P relationship generally increases during negative NDVI anomalies, in arid and semi-arid catchments, this response *slightly* decreases (Figure 2b). This decrease could *partially* be explained by reduced *hydrological* connectivity among bare patches *(Jaeger et al., 2014)* and increased soil evaporation (Guardiola-Claramonte et al. 2011). *However, these processes are highly dependent on the type, timing, and duration of drought, as well as catchment-specific characteristics (Goodwell et al., 2018; Liu et al., 2024), making generalizations challenging. Furthermore, we acknowledge that reduced transpiration, typically associated with negative NDVI anomalies, may also take place (Johnson et al., 2009). The interplay between these processes likely drives the observed variability, underscoring the need for caution when interpreting these results.*

9.  Section 4: The catchment memory to past conditions is heavily dependent on geology. Could the authors find a link between their results and geological characteristics?

> • We agree with the reviewer about the important role of geological characteristics in influencing changes in catchment responses to precipitation due to meteorological and hydrological anomalies. In this study, we specifically investigated soil type characteristics, as detailed in lines [233–236] of the manuscript. The results of this analysis are presented in Figure S9 of the Supplementary Information and discussed in lines [389–393]. Our findings indicate for instance that catchment response to precipitation in arid and equatorial sandy catchments is significantly influenced by soil moisture drought, while hydrological drought plays a key role in warm temperate catchments with both clay and sandy soils.

**References**

Anderson, B. J., Brunner, M. I., Slater, L. J., & Dadson, S. J. (2023). *Elasticity curves describe streamflow sensitivity to precipitation across the entire flow distribution*. https://doi.org/10.5194/hess-2022-407

Beck, H. E., Vergopolan, N., Pan, M., Levizzani, V., Van Dijk, A. I. J. M., Weedon, G. P., Brocca, L., Pappenberger, F., Huffman, G. J., & Wood, E. F. (2017). Global-scale evaluation of 22 precipitation datasets using gauge observations and hydrological modeling. *Hydrology and Earth System Sciences*, *21*(12). https://doi.org/10.5194/hess-21-6201-2017

Beck, H. E., Wood, E. F., Pan, M., Fisher, C. K., Miralles, D. G., Van Dijk, A. I. J. M., McVicar, T. R., & Adler, R. F. (2019). MSWep v2 Global 3-hourly 0.1° precipitation: Methodology and quantitative assessment. *Bulletin of the American Meteorological Society*, *100*(3). https://doi.org/10.1175/BAMS-D-17-0138.1

Bhanja, S. N., Mukherjee, A., Saha, D., Velicogna, I., & Famiglietti, J. S. (2016). Validation of GRACE based groundwater storage anomaly using in-situ groundwater level measurements in India. *Journal of Hydrology*, *543*. https://doi.org/10.1016/j.jhydrol.2016.10.042

Creed, I. F., Spargo, A. T., Jones, J. A., Buttle, J. M., Adams, M. B., Beall, F. D., Booth, E. G., Campbell, J. L., Clow, D., Elder, K., Green, M. B., Grimm, N. B., Miniat, C., Ramlal, P., Saha, A., Sebestyen, S., Spittlehouse, D., Sterling, S., Williams, M. W., … Yao, H. (2014). Changing forest water yields in response to climate warming: Results from long-term experimental watershed sites across North America. *Global Change Biology*, *20*(10). https://doi.org/10.1111/gcb.12615

Garreaud, R. D., Alvarez-Garreton, C., Barichivich, J., Pablo Boisier, J., Christie, D., Galleguillos, M., LeQuesne, C., McPhee, J., & Zambrano-Bigiarini, M. (2017). The 2010-2015 megadrought in central Chile: Impacts on regional hydroclimate and vegetation. *Hydrology and Earth System Sciences*, *21*(12). https://doi.org/10.5194/hess-21-6307-2017

Gebrechorkos, S. H., Leyland, J., Dadson, S. J., Cohen, S., Slater, L., Wortmann, M., Ashworth, P. J., Bennett, G. L., Boothroyd, R., Cloke, H., Delorme, P., Griffith, H., Hardy, R., Hawker, L., McLelland, S., Neal, J., Nicholas, A., Tatem, A. J., Vahidi, E., … Darby, S. E. (2024). Global-scale evaluation of precipitation datasets for hydrological modelling. *Hydrology and Earth System Sciences*, *28*(14). https://doi.org/10.5194/hess-28-3099-2024

Helman, D., Lensky, I. M., Yakir, D., & Osem, Y. (2017). Forests growing under dry conditions have higher hydrological resilience to drought than do more humid forests. *Global Change Biology*, *23*(7). https://doi.org/10.1111/gcb.13551

Holm, A. M. R., Cridland, S. W., & Roderick, M. L. (2003). The use of time-integrated NOAA NDVI data and rainfall to assess landscape degradation in the arid shrubland of Western Australia. *Remote Sensing of Environment*, *85*(2). https://doi.org/10.1016/S0034-4257(02)00199-2

Jaeger, K. L., Olden, J. D., & Pelland, N. A. (2014). Climate change poised to threaten hydrologic connectivity and endemic fishes in dryland streams. *Proceedings of the National Academy of Sciences of the United States of America*, *111*(38). https://doi.org/10.1073/pnas.1320890111

Mazzoleni, M., Brandimarte, L., & Amaranto, A. (2019). Evaluating precipitation datasets for large-scale distributed hydrological modelling. *Journal of Hydrology*, *578*. https://doi.org/10.1016/j.jhydrol.2019.124076

Nicholson, S. E., Davenport, M. L., & Malo, A. R. (1990). A comparison of the vegetation response to rainfall in the Sahel and East Africa, using normalized difference vegetation index from NOAA AVHRR. *Climatic Change*, *17*(2–3). https://doi.org/10.1007/BF00138369

Sankarasubramanian, A., Vogel, R. M., & Limbrunner, J. F. (2001). Climate elasticity of streamflow in the United States. *Water Resources Research*, *37*(6). https://doi.org/10.1029/2000WR900330

Satgé, F., Ruelland, D., Bonnet, M. P., Molina, J., & Pillco, R. (2019). Consistency of satellite-based precipitation products in space and over time compared with gauge observations and snow- hydrological modelling in the Lake Titicaca region. *Hydrology and Earth System Sciences*, *23*(1). https://doi.org/10.5194/hess-23-595-2019

Schaake, J. C. (1990). From Climate to Flow. In *Climate Change and U.S. Water Resources*.

Schmidt, R., Flechtner, F., Meyer, U., Neumayer, K. H., Dahle, C., König, R., & Kusche, J. (2008). Hydrological signals observed by the GRACE satellites. In *Surveys in Geophysics* (Vol. 29, Issues 4–5). https://doi.org/10.1007/s10712-008-9033-3

Van Loon, A. F., Kchouk, S., Matanó, A., Tootoonchi, F., Alvarez-Garreton, C., Hassaballah, K. E. A., Wu, M., Wens, M. L. K., Shyrokaya, A., Ridolfi, E., Biella, R., Nagavciuc, V., Barendrecht, M. H., Bastos, A., Cavalcante, L., de Vries, F. T., Garcia, M., Mård, J., Streefkerk, I. N., … Werner, M. (2024). *Review article: Drought as a continuum: memory effects in interlinked hydrological, ecological, and social systems*. https://doi.org/10.5194/egusphere-2024-421

Van Loon, A. F., Kumar, R., & Mishra, V. (2017). Testing the use of standardised indices and GRACE satellite data to estimate the European 2015 groundwater drought in near-real time. *Hydrology and Earth System Sciences*, *21*(4). https://doi.org/10.5194/hess-21-1947-2017

Wang, H., Xiang, L., Steffen, H., Wu, P., Jiang, L., Shen, Q., Li, Z., & Hayashi, M. (2022). GRACE-based estimates of groundwater variations over North America from 2002 to 2017. *Geodesy and Geodynamics*, *13*(1). https://doi.org/10.1016/j.geog.2021.10.003

Xue, B., Wang, G., Xiao, J., Helman, D., Sun, W., Wang, J., & Liu, T. (2020). Global convergence but regional disparity in the hydrological resilience of ecosystems and watersheds to drought. *Journal of Hydrology*, *591*. https://doi.org/10.1016/j.jhydrol.2020.125589

Zhang, Y., Viglione, A., & Blöschl, G. (2022). Temporal Scaling of Streamflow Elasticity to Precipitation: A Global Analysis. *Water Resources Research*, *58*(1). https://doi.org/10.1029/2021WR030601

---

## Author Comment (AC3)

**Drought decreases streamflow response to precipitation especially in arid regions**

**Reviewer 2**

The authors study the changing streamflow dynamics of catchments in response to changing precipitation. Disentangling the influence of climatic/weather and catchment properties on the streamflow response is rather complicated. This is regularly found in the efforts to explain the variability of streamflow elasticity or flow duration curve properties between catchments. Both typically end up with a mixture of explanatory variables. The current manuscript is well written and technical solid – as far as I can tell. However, like reviewer 1, I have some questions regarding definitions and the robustness of the results, given those definitions. I also have some questions regarding correlation versus causation.

- We sincerely thank the reviewer for their comments and are very pleased that they found the manuscript well written and appreciate our efforts in developing a solid method. The reviewer provides constructive feedback and suggestions, which we will address in the revised manuscript. Below, we summarize the changes we will make in response to these comments. Our responses are shown in blue, the revised text is shown in *italics*, and line numbers mentioned in this response refer to the current version of the manuscript and they are indicated within brackets [xx].

[1] As the other reviewer also states one relevant question is the definition of "streamflow sensitivity to precipitation". Is the streamflow sensitivity to precipitation well defined when we use the ratio between annual streamflow and precipitation? The mathematical definition of sensitivity is the change in the output due to variations in the input of a system or model. Plots of Q versus P do not quite capture this definition because they only look at the response of the system to the input, without consideration of what state the system was in. The latter is capture by the idea of streamflow elasticity where one quantifies the change in streamflow due to the change in precipitation (or something else) from year to year. Elasticity is notoriously difficult to explain (or regionalize) while the Q-P relationship is often rather stable. Given that this type of analysis was part of Anderson et al. (2024, HESS, "Elasticity curves describe streamflow sensitivity to precipitation across the entire flow distribution"), I think it would be very good if the authors were to make the connection and discuss how their definition in this manuscript differs from previous work (incl. some of the authors) and what consequences the changing definition has.

- The reviewer is correct in pointing out that the ratio between streamflow and precipitation does not provide information about the sensitivity of streamflow to precipitation but rather information on the yearly streamflow response to precipitation (annual catchment response to precipitation). Therefore the use of the terminology: "streamflow sensitivity to precipitation" is misleading and we will change it to: "streamflow response to precipitation". Further, we agree about the need to better define what the yearly ratio between streamflow and precipitation represent and to

compare it to other metrics used in the literature (e.g. elasticity). The added text will be for instance:

> [74] Here, we analysed the temporal dynamics of the *annual* streamflow *response* to precipitation (computed as the ratio between annual streamflow and precipitation) in approximately 5000 catchments across the world. *The annual Q-P ratio indicates the fraction of precipitation that is converted into streamflow, providing insights into the catchment's water balance.*
>
> [109] We then computed yearly streamflow-to-precipitation (Q-P) ratio timeseries for each catchment. *This measure represents the annual runoff ratio and is dynamically influenced by climatic and hydrological conditions. By considering an annual timescale, the ratio inherently accounts for evapotranspiration and storage processes within the catchment. However, it is important to note that, first, since the ratio is a lumped representation of these processes, it does not separate individual contributions. Second, in some catchments, storage processes extend beyond a single year, which may influence the annual runoff ratio. This metric differs from other metrics such as elasticity (Anderson et al., 2023; Sankarasubramanian et al., 2001; Zhang et al., 2022). While the annual runoff ratio provides an average measure of how much precipitation contributes to streamflow in a given year, elasticity tells us how streamflow reacts to changes in precipitation (Schaake, 1990).*

[2] My second larger point is about the difference between correlation and causation. The authors work here, and many of the papers cited, use correlation to infer causation. While I fully agree with the type of analysis, I think that it would be good to at least discuss somewhat whether correlation can be used here to infer causation. This also includes the discussion use of some of the references. One example is in lines 487ff. where the authors state: "This decrease could be explained by reduced connectivity among bare patches (Urgeghe et al. 2010)". The Urgeghe et al. study runs a model for a design storm and varies vegetation patches to show their role for runoff behaviour during the design storm. I find it quite a stretch to use this reference in support of long-term catchment water balance behaviour. The authors need to at least explain why they think this connection is valid. The second part of the sentence in lines 487ff. is "and increased soil evaporation due to an increase in solar radiation reaching the ground (Guardiola-Claramonte et al. 2011)." Isn't the latter than coinciding with reduced transpiration? Is the reduction in transpiration not larger than the added soil evaporation (given the deeper capture of moisture through roots)?

This is just an example where I think where the authors could expand their discussion and argument. I just given an example, reflective of the wider discussion section. It would in general be good if the authors were a bit more explicit why the references cited are transferrable to their situation.

- We thank the reviewer for this comment and fully agree on the need for a more careful use of references and caution in inferring causation from correlation. In the discussion

section, our intention was to assess whether our results align with findings from other studies and to explore how these findings have been explained in terms of underlying processes. Upon revisiting the references, we acknowledge that the use of Urgeghe et al. (2010) may indeed be an overextension in this context. As well as the use of Garreaud et al., 2017, so we will delete it.

Additionally, upon further review of the literature, we recognize that drought in arid regions does not always lead to a decrease in hydrological connectivity. For example, (Ruddell & Kumar, 2009) highlight cases where connectivity decreases, whereas other studies (Goodwell et al., 2018; Liu et al., 2024) document increases in connectivity under certain conditions. These differences depend on factors such as the type, timing, and duration of drought, as well as vegetation type and other catchment characteristics, making generalizations difficult.

We also agree with the reviewer that negative NDVI anomalies typically reflect reduced transpiration compared to the system's baseline (Johnson et al., 2009). However, variations in transpiration and soil evaporation during negative NDVI differ across systems (Dijke et al., 2019; Lawrence et al., 2007).

We propose to revise the discussion as follows:

[485] Spatial differences can also be found in the influence of negative NDVI anomalies on the Q-P relationship, *though the overall influence remains small (less than 5%)*. While the response of the Q-P relationship generally increases during negative NDVI anomalies, in arid and semi-arid catchments, this response *slightly* decreases (Figure 2b). This decrease could *partially* be explained by reduced *hydrological* connectivity among bare patches *(Jaeger et al., 2014)* and increased soil evaporation (Guardiola-Claramonte et al. 2011). *However, these processes are highly dependent on the type, timing, and duration of drought, as well as catchment-specific characteristics (Goodwell et al., 2018; Liu et al., 2024), making generalizations challenging. Furthermore, we acknowledge that reduced transpiration, typically associated with negative NDVI anomalies, may also take place (Johnson et al., 2009). The interplay between these processes likely drives the observed variability, underscoring the need for caution when interpreting these results.*

[3] Influence of length of about 30 time steps on the robustness of the stationarity test? It seems like a very short time span for such tests. And given the widely discussed limitations of using statistical significance for justification. This is not a criticism of the approach, but a question of how one can assure robustness of the results?

- We agree with the reviewer that the robustness of the ADF test can be influenced by the length of the time series. With a short time series (~30 points), the ADF test is more likely

to misclassify a stationary series as non-stationary. The test might indeed not find enough evidence to reject the null hypothesis (H0: "the series is non-stationary").

In our study, the identified non-stationary time series are used to investigate possible shifts in streamflow response to precipitation, transitioning from one steady state to another. To ensure the robustness of our results and confirm that these time series present a significant shift and, hence are truly non-stationary, we performed in our study additional tests (see section 2.4 of the manuscript). Indeed, we examined whether these time series exhibit a linear, curvilinear, or abrupt (characterised by sudden changes) trend. Abrupt changes were specifically tested using a threshold regression approach (see lines 256–267). The best-fitting trend was then identified using the Akaike Information Criterion (AIC). Further, to account for potential uncertainty due to short time series and data noise, we bootstrapped each time series 100 times without replacement and compared the model results across iterations (lines 268–276). Finally, to further increase confidence in detecting step changes, we applied a series of restrictive criteria (see lines 268–299). Through these additional steps, we increased robustness of our results and hence that the time series used in our step-change analysis are non-stationary, as they exhibit a significant shift in their structure.

[4] Temporal connectivity of drought events? Is there relevance to the temporal sequence of drought periods for this analysis. Even though I appreciate that the short time series might make this difficult to study.

- As the reviewer suggests, we could not fully account for the temporal connectivity of drought events due to the relatively short time series used in the analysis. However, we partially explored temporal connectivity by analysing the influence of droughts occurring in the preceding year. These influences were incorporated into our analysis through Equation 1. We wrote a reflection on this in the discussion:

  > [568] *Although drought is a continuum, with temporal connectivity between events (Van Loon et al., 2024), our analysis treats droughts as independent events, summarizing their characteristics at a yearly scale to facilitate comparison with the yearly ratio of Q to P. We only partially accounted for drought connectivity by incorporating drought characteristics from the preceding year into our analysis. However, their influence was minimal (less than 5%), with meteorological drought showing a slightly higher influence compared to other drought types.*

[5] How relevant are conclusions that show differences of 2-3%? This should be quite below the amount of uncertainty one would expect in precipitation and/or streamflow observations even in good circumstances.

- If the reviewer is referring to cases where our results differ by 2–3%, such as the 30% and 27% influence of hydrological and soil moisture drought on the Q-P ratio (lines 344–345), we agree that this difference is minimal and likely irrelevant given the inherent

uncertainties in precipitation and streamflow observations. We will explicitly acknowledge this in the limitations of our study.

[572] *The accuracy of the percentage values representing the influence of a certain drought type on the yearly Q-P ratio is affected by uncertainties in precipitation and streamflow observations. Although these percentage values are not exact due to observational uncertainties, the relative magnitudes provides meaningful information, allowing us to identify which drought types have the strongest influence on the Q-P ratio.*

If, instead, the reviewer is questioning the relevance of drought types with a low influence (~2–3%) on the Q-P ratio, such as NDVI anomalies (line 309), we argue that the broader finding remains valid despite observational uncertainties. A relatively small influence suggests that this specific drought type has minimal impact on catchment response. In contrast to the 20–30% changes observed for other drought types, this lower effect may indicate that these catchments are more resilient to changes associated, for instance, with NDVI anomalies.

**References**

Anderson, B. J., Brunner, M. I., Slater, L. J., & Dadson, S. J. (2023). *Elasticity curves describe streamflow sensitivity to precipitation across the entire flow distribution*. https://doi.org/10.5194/hess-2022-407

Dijke, A., Mallick, K., Teuling, A., Schlerf, M., Machwitz, M., Hassler, S., Blume, T., & Herold, M. (2019). Does the Normalized Difference Vegetation Index explain spatial and temporal variability in sap velocity in temperate forest ecosystems? *Hydrology and Earth System Sciences*, *23*(4). https://doi.org/10.5194/hess-23-2077-2019

Garreaud, R. D., Alvarez-Garreton, C., Barichivich, J., Pablo Boisier, J., Christie, D., Galleguillos, M., LeQuesne, C., McPhee, J., & Zambrano-Bigiarini, M. (2017). The 2010-2015 megadrought in central Chile: Impacts on regional hydroclimate and vegetation. *Hydrology and Earth System Sciences*, *21*(12). https://doi.org/10.5194/hess-21-6307-2017

Goodwell, A. E., Kumar, P., Fellows, A. W., & Flerchinger, G. N. (2018). Dynamic process connectivity explains ecohydrologic responses to rainfall pulses and drought. *Proceedings of the National Academy of Sciences of the United States of America*, *115*(37). https://doi.org/10.1073/pnas.1800236115

Jaeger, K. L., Olden, J. D., & Pelland, N. A. (2014). Climate change poised to threaten hydrologic connectivity and endemic fishes in dryland streams. *Proceedings of the National Academy of Sciences of the United States of America*, *111*(38). https://doi.org/10.1073/pnas.1320890111

Johnson, G. L., Sinclair, T. R., & Kenworthy, K. (2009). Transpiration and normalized difference vegetation index response of seashore paspalum to soil drying. *HortScience*, *44*(7). https://doi.org/10.21273/hortsci.44.7.2046

Lawrence, D. M., Thornton, P. E., Oleson, K. W., & Bonan, G. B. (2007). The partitioning of evapotranspiration into transpiration, soil evaporation, and canopy evaporation in a GCM: Impacts on land-atmosphere interaction. *Journal of Hydrometeorology*, *8*(4). https://doi.org/10.1175/JHM596.1

Liu, C., Chen, Y., Huang, W., Fang, G., Li, Z., Zhu, C., & Liu, Y. (2024). Climate warming positively affects hydrological connectivity of typical inland river in arid Central Asia. *Npj Climate and Atmospheric Science*, *7*(1). https://doi.org/10.1038/s41612-024-00800-4

Ruddell, B. L., & Kumar, P. (2009). Ecohydrologic process networks: 1. Identification. *Water Resources Research*, *45*(3). https://doi.org/10.1029/2008WR007279

Sankarasubramanian, A., Vogel, R. M., & Limbrunner, J. F. (2001). Climate elasticity of streamflow in the United States. *Water Resources Research*, *37*(6). https://doi.org/10.1029/2000WR900330

Schaake, J. C. (1990). From Climate to Flow. In *Climate Change and U.S. Water Resources*.

Van Loon, A. F., Kchouk, S., Matanó, A., Tootoonchi, F., Alvarez-Garreton, C., Hassaballah, K. E. A., Wu, M., Wens, M. L. K., Shyrokaya, A., Ridolfi, E., Biella, R., Nagavciuc, V., Barendrecht, M. H., Bastos, A., Cavalcante, L., de Vries, F. T., Garcia, M., Mård, J., Streefkerk, I. N., … Werner, M. (2024). *Review article: Drought as a continuum: memory effects in interlinked hydrological, ecological, and social systems*. https://doi.org/10.5194/egusphere-2024-421

Zhang, Y., Viglione, A., & Blöschl, G. (2022). Temporal Scaling of Streamflow Elasticity to Precipitation: A Global Analysis. *Water Resources Research*, *58*(1). https://doi.org/10.1029/2021WR030601